# cGAS-like receptors sense RNA and control 3′2′-cGAMP signalling in *Drosophila*

Kailey M. Slavik[1,2], Benjamin R. Morehouse[1,2], Adelyn E. Ragucci[1,2], Wen Zhou[1,2,9], Xianlong Ai[3], Yuqiang Chen[3], Lihua Li[3], Ziming Wei[3], Heike Bähre[4], Martin König[4], Roland Seifert[4,5], Amy S. Y. Lee[2,6], Hua Cai[3], Jean-Luc Imler[3,7] & Philip J. Kranzusch[1,2,8 ✉]

Cyclic GMP–AMP synthase (cGAS) is a cytosolic DNA sensor that produces the second messenger cG[2′–5′]pA[3′–5′]p (2′3′-cGAMP) and controls activation of innate immunity in mammalian cells[1–5]. Animal genomes typically encode multiple proteins with predicted homology to cGAS[6–10], but the function of these uncharacterized enzymes is unknown. Here we show that cGAS-like receptors (cGLRs) are innate immune sensors that are capable of recognizing divergent molecular patterns and catalysing synthesis of distinct nucleotide second messenger signals. Crystal structures of human and insect cGLRs reveal a nucleotidyltransferase signalling core shared with cGAS and a diversified primary ligand-binding surface modified with notable insertions and deletions. We demonstrate that surface remodelling of cGLRs enables altered ligand specificity and used a forward biochemical screen to identify cGLR1 as a double-stranded RNA sensor in the model organism *Drosophila melanogaster*. We show that RNA recognition activates *Drosophila* cGLR1 to synthesize the novel product cG[3′–5′]pA[2′–5′]p (3′2′-cGAMP). A crystal structure of *Drosophila* stimulator of interferon genes (dSTING) in complex with 3′2′-cGAMP explains selective isomer recognition, and 3′2′-cGAMP induces an enhanced antiviral state in vivo that protects from viral infection. Similar to radiation of Toll-like receptors in pathogen immunity, our results establish cGLRs as a diverse family of metazoan pattern recognition receptors.

To define the function of cGAS-like enzymes in animals, we screened predicted cGAS homologues for suitability in structural analysis and determined a 2.4 Å crystal structure of the human protein MB21D2 (hMB21D2; encoded by *C3orf59*) and a 1.6 Å crystal structure of a protein from the beetle species *Tribolium castaneum* (GenBank XP_969398.1) (Supplementary Table 1). Despite divergence in the primary sequence, the hMB21D2 and *T. castaneum* XP_969398.1 structures each reveal close homology to human cGAS with a shared bi-lobed architecture, a caged nucleotidyltransferase core, a Gly-[Gly/Ser] activation loop and a putative catalytic triad (Fig. 1a, Extended Data Fig. 1). In human cGAS, the primary ligand-binding surface is a long groove on the back of the enzyme formed by the α-helix spine and a Zn-ribbon motif that is essential for recognition of double-stranded DNA[3,11–14]. A conserved groove is present in both the hMB21D2 and the *T. castaneum* XP_969398.1 structures (Fig. 1a), but is notably distinguished by the absence of a Zn-ribbon and the insertion of a C-terminal α-helix in hMB21D2 (Fig. 1b). We hypothesized that the remodelling of this groove controls the detection of distinct ligands. The hMB21D2 surface is overall neutral with no obvious capacity to bind nucleic acids, and no enzymatic activity was detected with a panel of potential activating ligands (Extended Data Fig. 1d, e). In contrast to hMB21D2, the surface of *T. castaneum*

XP_969398.1 shares highly conserved basic residues with human cGAS (Fig. 1a) and we therefore tested this enzyme with candidate DNA and RNA ligands. We observed that *T. castaneum* XP_969398.1 is activated to synthesize a nucleotide product upon recognition of double-stranded RNA (dsRNA) (Fig. 1c). Despite exhibiting a clear difference in ligand specificity, analysis of all structures in the Protein Data Bank confirmed that *T. castaneum* XP_969398.1 is a close homologue of mammalian cGAS and is distinct from previously characterized RNA sensors including oligoadenylate synthase 1 (ref. [15]) (Extended Data Fig. 1f). Together, these results establish the existence of cGLRs in animals and demonstrate that remodelling of a primary ligand-binding surface enables the recognition of divergent molecular patterns.

To identify additional cGLRs that respond to dsRNA, we used the *T. castaneum* cGLR (*Tc*-cGLR) sequence to search for predicted cGAS homologues in species related to the model organism *D. melanogaster*. We identified 153 cGLR genes across 42 species in the order Diptera, which cluster into distinct clades designated 1–5 (Fig. 2a, Supplementary Table 2). *Drosophila* encode a remarkable number of cGLR genes, with individual species predicted to have between three and seven enzymes (Extended Data Fig. 2a). In a systematic biochemical screen, we purified and tested 53 recombinant cGLR proteins and identified active

[1]Department of Microbiology, Harvard Medical School, Boston, MA, USA. [2]Department of Cancer Immunology and Virology, Dana-Farber Cancer Institute, Boston, MA, USA. [3]Sino-French Hoffmann Institute, State Key Laboratory of Respiratory Disease, School of Basic Medical Science, Guangzhou Medical University, Guangzhou, China. [4]Research Core Unit Metabolomics, Hannover Medical School, Hannover, Germany. [5]Hannover Medical School, Institute of Pharmacology, Hannover, Germany. [6]Department of Cell Biology, Harvard Medical School, Boston, MA, USA. [7]Université de Strasbourg, CNRS UPR9022, Strasbourg, France. [8]Parker Institute for Cancer Immunotherapy at Dana-Farber Cancer Institute, Boston, MA, USA. [9]Present address: School of Life Sciences, Southern University of Science and Technology, Guangdong, China. ✉e-mail: philip_kranzusch@dfci.harvard.edu

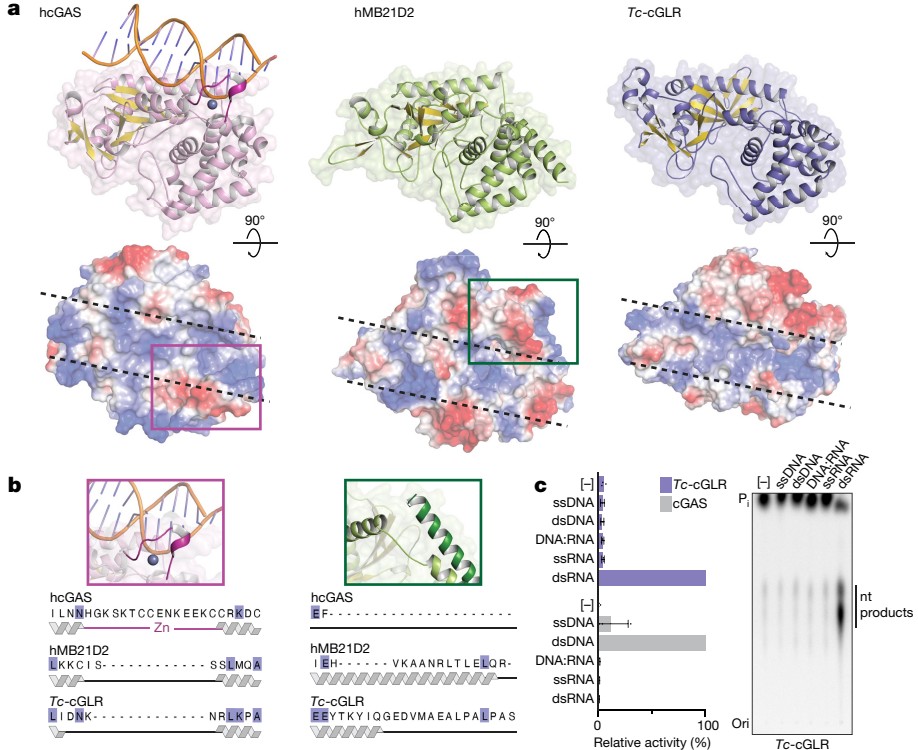

**Fig. 1 | Structural remodelling in animal cGLRs enables divergent pattern recognition. a**, Crystal structures and surface electrostatics of hMB21D2 and *Tc*-cGLR. Structural comparison with the human cGAS (hcGAS)–DNA complex (Protein Data Bank (PDB): 6CTA)[14] reveals that cGLRs have a conserved architecture with a nucleotidyltransferase signalling core and a shared primary ligand-binding surface (dashed lines). The purple and green boxes indicate cutaways in **b**. **b**, Zoomed-in cutaways highlighting structural insertions and deletions unique to each cGLR. hMB21D2 and *Tc*-cGLR lack the Zn-ribbon motif present in cGAS (left) and hMB21D2 contains a C-terminal α-helix extension

that contacts the central 'spine' helix (right). Alterations in the predicted ligand-binding surfaces suggest individual cGLRs are remodelled to recognize different molecular patterns. **c**, Thin-layer chromatography analysis and quantification of *Tc*-cGLR reactions in the presence of nucleic acid ligands. *Tc*-cGLR is specifically activated by dsRNA recognition to synthesize a nucleotide (nt) product. Data are relative to maximum activity and represent the mean ± s.e.m. for *n* = 3 independent experiments. Ori, origin; P$_i$, inorganic phosphate; ss, single-stranded.

enzymes from the species *Lucilia cuprina*, *Drosophila eugracilis*, *Drosophila erecta* and *Drosophila simulans* (Extended Data Figs. 2b–f, 3a). Similar to *Tc*-cGLR, each active Diptera enzyme specifically responded to dsRNA, indicating that cGLR-based recognition of RNA is conserved across diverse insect species (Fig. 2b, Extended Data Fig. 4a).

The *D. simulans* enzyme identified in our screen shares 91% sequence identity with the protein product of the *D. melanogaster* gene *CG12970* (GenBank NP_788360.2). Analysis of recombinant *D. melanogaster* CG12970 protein revealed that it also synthesizes a nucleotide product specifically in the presence of dsRNA and we therefore named this gene *cGAS-like Receptor 1* (*Dm-cGLR1*) (Fig. 2c). To understand how dsRNA activates *Drosophila* cGLR1, we analysed the molecular determinants for enzymatic activity in vitro. We observed that *D. simulans* cGLR1 (*Ds*-cGLR1) and *Dm*-cGLR1 recognize dsRNAs longer than 30 bp with no preference for 5′ RNA phosphorylation (Fig. 2d, Extended Data Fig. 4b, c). Notably, activation of *Ds*-cGLR1 and *Dm*-cGLR1 requires dsRNA ligands that exceed the length of 21–23-bp RNA molecules commonly produced during RNA interference in *Drosophila*, suggesting specific avoidance of self-recognition[16,17]. Similar to the formation of condensates observed with human cGAS recognition of dsDNA[18], *Ds*-cGLR1 selectively binds to dsRNA and forms a higher-order complex that is dependent on the length of dsRNA (Extended Data Fig. 5). Ectopic expression of *Dm*-cGLR1 or *Ds*-cGLR1 in human cells demonstrated that cGLR1 activity is sufficient to enable cellular dsRNA sensing and drive activation of a STING-dependent immune response (Fig. 2e, Extended Data Figs. 3f, 4e). *Dm*-cGLR1 and *Ds*-cGLR1 signalling in cells required dsRNA stimulation, and mutations to the enzyme catalytic site disrupted downstream activation of STING (Fig. 2e, Extended Data Fig. 3f).

To understand how *Drosophila* cGLR1 engages dsRNA, we modelled interactions using the *Tc*-cGLR and human cGAS–DNA structures as a template[14] and observed that charge-swap mutations to the conserved basic ligand-binding surface disrupted product synthesis in vitro and STING signalling in cells (Fig. 2e, Extended Data Fig. 3c–f). Together, these data demonstrate that insect cGLRs and human cGAS use a shared mechanism of ligand detection and reveal that *Dm*-cGLR1 can function as a foreign RNA sensor.

A role in sensing long dsRNA suggests that the function of *Dm*-cGLR1 is to control a downstream immune response in *Drosophila*. In human cells, cGAS synthesizes the nucleotide second messenger 2′3′-cGAMP, which contains a non-canonical 2′–5′ phosphodiester linkage that is required for potent activation of immune signalling[2–5]. To determine how *Dm*-cGLR1 controls cellular signalling, we purified the nucleotide reaction product for direct comparison to 2′3′-cGAMP. The *Dm*-cGLR1 product exhibited a C18 chromatography migration profile distinct from 2′3′-cGAMP and all previously known naturally occurring cyclic dinucleotide (CDN) signals (Fig. 3a, Extended Data Fig. 6a). Production of this nucleotide signal was conserved in Diptera with *Ds*-cGLR1, *Lc*-cGLR and *Deu*-cGLR reactions, each synthesizing the same major reaction product (Extended Data Fig. 6a). Using nucleobase-specific labelling and nuclease digestion of the *Dm*-cGLR1 product, we observed a 3′–5′ linkage connected to an adenosine phosphate and a protected 2′–5′ linkage connected to a guanosine phosphate, indicating a mixed-linkage cyclic GMP–AMP species (Fig. 3b). We verified these findings with comparative high-performance liquid chromatography and tandem mass spectrometry profiling against a chemically synthesized standard, and confirmed that the shared Diptera cGLR

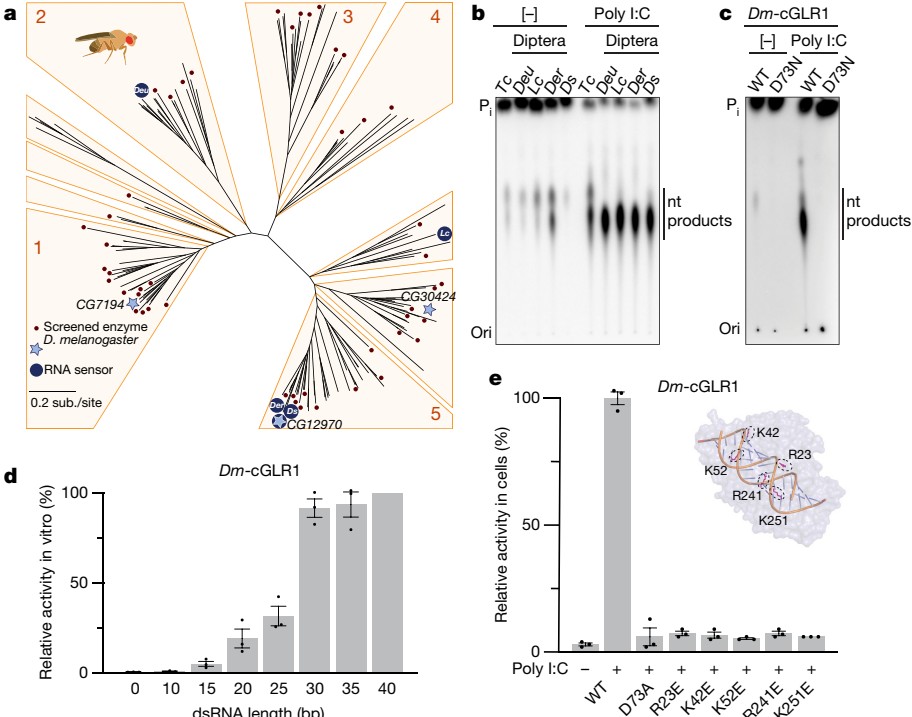

**Fig. 2 | _Drosophila_ cGLR1 senses long dsRNA. a**, Phylogeny representing 153 Diptera cGLR genes clustered into clades 1–5 (less than 30% sequence identity between clades). Forty-one of forty-two analysed Diptera species encode enzymes in clade 5, including _D. melanogaster_ CG12970 (cGLR1) and CG30424. Enzymes analysed by forward biochemical screen (red dot) and identified as dsRNA-sensing cGLRs (blue circle) are denoted. **b**, Diptera cGLRs identified in the screen are activated to form a nucleotide product by the dsRNA mimic poly I:C. **c**, Mutation to the _Dm_-cGLR1 active site ablates enzymatic activity. Data in **b** and **c** are representative of _n_ = 3 independent experiments. WT, wild type. **d**, _Dm_-cGLR1 in vitro activity was monitored in the presence of a panel of

dsRNAs and quantified relative to 40 bp dsRNA. Data are the mean ± s.e.m. of _n_ = 3 independent experiments. **e**, Analysis of _Dm_-cGLR1 activity in human cells using mammalian STING and IFNβ reporter induction, quantified relative to WT activity. _Dm_-cGLR1 signalling in cells is dependent on stimulation of dsRNA and mutation of the catalytic site, or the predicted ligand-binding residues ablates activity. Data are mean ± s.e.m. of _n_ = 3 technical replicates and representative of _n_ = 3 independent experiments. The inset shows a model of the _Tc_-cGLR–dsRNA complex based on the hcGAS–dsDNA structure (PDB: 6CTA)[14] used to predict the _Dm_-cGLR1 ligand-binding residues R23, K42, K52, R241 and K251.

product is the novel isomer 3′2′-cGAMP (Fig. 3a, b, Extended Data Fig. 6a, b).

_Dm_-cGLR1 synthesizes 3′2′-cGAMP in a two-step reaction through production of the linear intermediate pppA[2′–5′]pG and uses an opposite nucleobase reaction order compared with human cGAS[2,3,19] (Extended Data Fig. 7a). We next used mass spectrometry to analyse lysates expressing each recombinant dipteran cGLR from our screen. 3′2′-cGAMP was detected as a product of 15 cGLRs, including enzymes from each subgroup within clade 5 of the Diptera cGLR phylogeny (Extended Data Fig. 6c). cGLRs clustered within clade 5 collectively represent 41 species, suggesting widespread conservation of 3′2′-cGAMP signalling in Diptera. The beetle enzyme _Tc_-cGLR synthesizes 2′3′-cGAMP, supporting that 2′3′-cGAMP is an ancestral signalling molecule in metazoans and that 3′2′-cGAMP signalling is a recent adaptation in flies[8,20,21] (Fig. 3c, Extended Data Fig. 6a). Insect and mammalian viruses encode 2′3′-cGAMP-specific nucleases named poxins that allow evasion of cGAS–STING immune responses[22]. 3′2′-cGAMP was protected from cleavage by poxin (Extended Data Fig. 7b–d), indicating that an isomeric switch in the specificity of phosphodiester linkage endows _Drosophila_ with a signalling pathway resistant to a major form of viral immune evasion.

_Drosophila_ STING (dSTING) is known to function as a cyclic dinucleotide receptor in vivo[23–26], but an endogenous nucleotide second messenger has not been previously identified. We therefore developed an in vitro thermo-fluor binding assay to analyse dSTING recognition of specific CDNs. dSTING preferentially formed a thermostable complex with 3′2′-cGAMP and exhibited no detectable complex formation with 2′3′-cGAMP or other CDNs in vitro (Fig. 3d, Extended Data Fig. 8b,c).

Using direct delivery of CDNs to permeabilized cells, we confirmed that dSTING preferentially responds to 3′2′-cGAMP in the cellular environment (Extended Data Fig. 8d). To define the mechanism of selective 3′2′-cGAMP recognition, we next determined a 2.0 Å crystal structure of the _D. eugracilis_ STING (GenBank XP_017066673) CDN-binding domain in complex with 3′2′-cGAMP (Fig. 3e, Supplementary Table 1). dSTING adopts a highly conserved V-shaped homodimeric architecture with a deep central pocket that binds to 3′2′-cGAMP. The dSTING–3′2′-cGAMP structure reveals a tightly 'closed' conformation with dSTING protomers positioned 36 Å apart, similar to the closed conformation of human STING bound to 2′3′-cGAMP[5] (Extended Data Fig. 8e). Each nucleobase of 3′2′-cGAMP is stacked between dSTING Y164 and R234, and E257 specifically coordinates the 3′2′-cGAMP guanosine N2 position (Extended Data Fig. 8f). In human STING, high-affinity recognition of 2′3′-cGAMP requires readout of the 2′–5′ phosphodiester linkage by R232 in the β-strand lid[5]. In dSTING, the equivalent R229 makes no contact with either phosphodiester bond. Instead, R229 is repositioned to extend outwards from the ligand-binding pocket by the deletion of a single lid residue and the formation of a salt bridge with E267 on the opposing protomer, explaining the diminished affinity of dSTING for 2′3′-cGAMP (Fig. 3f, g). In addition, a key asparagine substitution, N159, in dSTING extends across the binding pocket to coordinate the adenosine 3′ OH in 3′2′-cGAMP and directly replaces the human STING S162 residue that contacts the guanosine 3′ OH in 2′3′-cGAMP (Fig. 3f, g). We tested a panel of dSTING-mutant proteins and confirmed that mutations to each coordinating residue disrupt the formation of the dSTING–3′2′-cGAMP complex (Extended Data Fig. 8i). The unique adaptations in the ligand-binding pocket observed in the

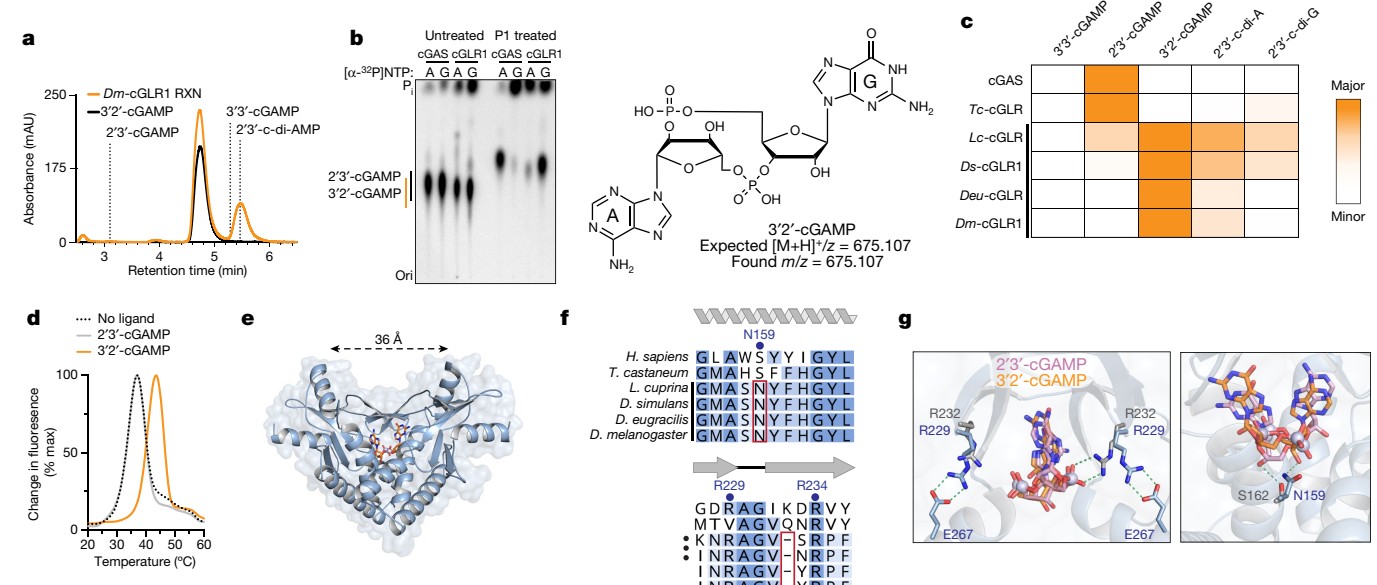

**Fig. 3 | Discovery of 3′2′-cGAMP as a metazoan nucleotide second messenger. a**, High-performance liquid chromatography (HPLC) analysis of the *Dm*-cGLR1 reaction (orange) and comparison with synthetic standards (black or dashed lines) demonstrates that *Dm*-cGLR1 synthesizes 3′2′-cGAMP as the major product. A minor *Dm*-cGLR1 reaction product is 2′3′-c-di-AMP (see also Extended Data Fig. 6a). **b**, Thin-layer chromatography analysis of mouse cGAS and *Dm*-cGLR1 reactions labelled with either α-³²P-ATP or α-³²P-GTP (indicated as [α-³²P]NTP) and treated as indicated. Pairwise labelling and nuclease P1 digestion verify that cGAS and *Dm*-cGLR1 synthesize distinct cGAMP isomers with opposite phosphodiester linkage specificities. Representative of *n* = 3 independent experiments. High-resolution mass spectrometry confirms the major Diptera cGLR product as 3′2′-cGAMP (see also Extended Data Fig. 6b). **c**, HPLC quantification of insect cGLR nucleotide products. 3′2′-cGAMP is the dominant product of each identified Diptera cGLR (denoted by a black line), and 2′3′-cGAMP is the dominant product of cGAS and *Tc*-cGLR. Data are the mean quantified product of *n* = 3 independent experiments. **d**, Thermal denaturation assay showing that dSTING selectively recognizes 3′2′-cGAMP (see also Extended Data Fig. 8b, c). Representative of *n* = 3 independent experiments. **e**, Crystal structure of the dSTING–3′2′-cGAMP complex reveals a tightly closed homodimer and an ordered β-strand lid, indicating high-affinity engagement of the endogenous *Drosophila* second messenger 3′2′-cGAMP. **f**, Alignment and conservation of the stem helix and β-strand lid in human and insect STING proteins. Critical ligand-binding residues (blue dot) and adaptations specific to Diptera (red outline) are denoted. **g**, Superposition of the dSTING–3′2′-cGAMP (blue–orange) complex and the human STING–2′3′-cGAMP (grey–pink) (PDB: 4KSY)[5] complex reveals that human STING readout of the 2′–5′ phosphodiester bond by R232 is absent in dSTING (left). Human STING S162 (grey) contacts the free 3′ OH of the guanosine base in 2′3′-cGAMP (pink). dSTING N159 (blue) extends across the ligand-binding pocket to contact the free 3′ OH of the adenosine base in 3′2′-cGAMP (orange) (right).

dSTING–3′2′-cGAMP structure are widely conserved in Diptera and together explain a mechanism for how specific 3′2′-cGAMP-dependent signalling drives the activation of dSTING.

To determine how *Dm*-cGLR1–3′2′-cGAMP–dSTING signalling controls immune responses in vivo, we next injected 3′2′-cGAMP into *D. melanogaster* to directly monitor the dSTING response. 3′2′-cGAMP potently induced the expression of *Sting* and three other *Sting*-regulated genes (*Srg*) in a dose-dependent manner (Fig. 4a, Extended Data Fig. 9). Notably, 3′2′-cGAMP-dependent signalling through dSTING was significantly more potent than the response triggered by injection of the bacterial CDN signal 3′3′-c-di-GMP (Fig. 4a, Extended Data Fig. 9e–k). Genetic mutations to *Sting* and the NF-κB homologue *Relish* ablated 3′2′-cGAMP-induced responses, demonstrating that signalling operates through a conserved dSTING–NF-κB pathway (Fig. 4a, Extended Data Fig. 9e–k). We challenged flies with viral infection and observed that 3′2′-cGAMP markedly suppressed the replication of two unrelated RNA viruses: *Drosophila* C virus (*Dicistroviridae*), a natural *Drosophila* pathogen, and vesicular stomatitis virus (*Rhabdoviridae*) (Fig. 4b, c, Extended Data Fig. 10a, b). 3′2′-cGAMP activation of antiviral immunity was strictly dependent on *Sting* and resulted in a response that significantly delayed pathogen-mediated mortality (Fig. 4b, c, Extended Data Fig. 10a, b). Direct comparison of the protective effects against *Drosophila* C virus infection showed that the endogenous signal 3′2′-cGAMP exhibited greater antiviral potency than 2′3′-cGAMP. 3′2′-cGAMP more robustly suppressed RNA viral loads and extended animal survival (Fig. 4d, Extended data Fig. 10c, d), revealing that the dSTING

antiviral signalling axis is preferentially activated by 3′2′-cGAMP in vivo. Together, these results demonstrate that 3′2′-cGAMP is an antiviral nucleotide second messenger in *D. melanogaster* and establish a cGLR–STING–NF-κB axis that protects animals from viral replication.

Along with cGAS recognition of dsDNA, the discovery of animal cGLR dsRNA sensors establishes a diverse class of pattern recognition receptors conserved throughout metazoans. Divergent structural homologues of cGAS in humans and insects demonstrate that cGLRs constitute a rapidly evolving family of proteins in which remodelling of a primary binding surface enables the detection of diverse ligands. Our mechanistic characterization of *Drosophila* cGLR1 activation shows that cGLRs function as direct sensors of pathogen-associated molecular patterns and synthesize distinct second messengers to control a conserved downstream signalling axis (Fig. 4e). *Drosophila* were previously thought to respond to foreign nucleic acid exclusively through RNA interference and direct cleavage of pathogen RNA[16,17]. *Drosophila* cGLR1 reveals a parallel signalling system for sensing dsRNA and directing an inducible immune response through dSTING. Synthesis of the second messenger 3′2′-cGAMP by *Drosophila* cGLR1 and selective recognition by dSTING demonstrates that metazoans use CDNs beyond 2′3′-cGAMP as endogenous second messengers and highlights the evolutionary plasticity of cGLR signalling. Our structural analysis also reveals that the human cGLR MB21D2 is competent for synthesis of nucleotide second messengers and has a remodelled ligand-binding groove that is probably adapted for detection of an unknown stimulus. Together with the known high frequency of hMB21D2 mutations in cancer[27,28],

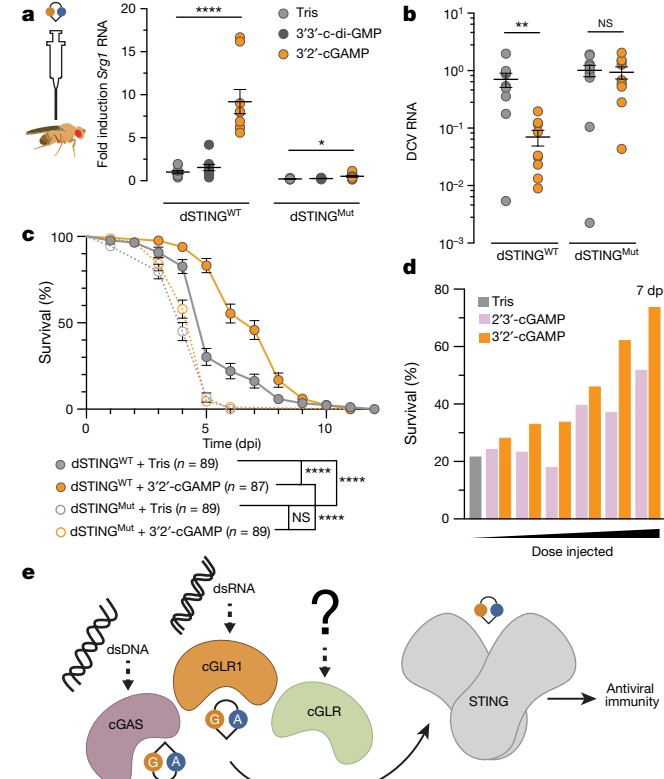

**Fig. 4 | 3′2′-cGAMP activates STING-dependent antiviral immunity in *Drosophila*. a**, Synthetic 3′2′-cGAMP or 3′3′-c-di-GMP was injected into the body cavity of flies and gene expression was measured after 24 h. *Sting-regulated gene 1 (Srg1)* RNA levels are shown as fold induction compared with buffer control in WT. dSTING^Mut is the RXN mutant, as previously characterized[23,26]. Data in **a** and **b** are mean ± s.e.m. of RNA levels measured relative to the control gene *Rpl32* from $n = 3$ independent experiments of $n = 6$ flies. The *P* values, calculated by unpaired *t*-test, were not significant (NS) unless otherwise noted; NS $P > 0.05$. ****$P < 0.001$, *$P = 0.0119$. **b**, Viral RNA loads 3 days after infection with *Drosophila* C virus (DCV) demonstrate significantly diminished viral replication in WT flies injected with 3′2′-cGAMP. **$P = 0.051$. **c**, Survival analysis of animals infected with DCV demonstrates that injection of 3′2′-cGAMP results in a *Sting*-dependent response that significantly delays mortality. Data are mean ± s.e.m. ****$P < 0.001$. Data in **c** and **d** are each from $n = 3$ independent experiments of $n = 30$ flies. dpi, days post-infection. **d**, Survival analysis directly comparing the effects of cGAMP isomers 7 days after DCV infection. 3′2′-cGAMP injection increases animal survival in a dose-dependent manner and confers greater protection than 2′3′-cGAMP (see also Extended Data Fig. 10d). **e**, Proposed model for cGLR–STING signalling. Upon recognition of distinct molecular patterns, animal cGLRs synthesize a nucleotide second messenger that activates antiviral immunity through STING.

these results support a more extensive role for cGLR signalling in human biology. The existence of multiple unique cGLRs encoded within a single species (Extended Data Fig. 2a) suggests a model in which the cGLR signalling scaffold is harnessed to detect several distinct stimuli. In support of this conclusion, Hartmann, Imler, Cai and colleagues have identified cGLR2 as a second functional cGLR in *Drosophila* and have demonstrated in vivo that cGLR1 and cGLR2 have discrete roles in *Drosophila* immunity[29]. Together, our results define cGLRs as receptors in animal cells that are capable of detecting diverse pathogen-associated molecular patterns and dictating response to the foreign environment.

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

## Methods

### Bioinformatics and dipteran cGLR sequence analysis

Building on previous analyses[6–10,30,31], animal cGAS homologues suitable for crystallography were identified using the amino acid sequences of human cGAS (hcGAS) and *D. melanogaster* CG7194 to seed a position-specific iterative BLAST (PSI-BLAST) search of the NCBI non-redundant protein database. The PSI-BLAST search was performed with an *E* value cut-off of 0.005 for inclusion into the next search round, BLOSUM62 scoring matrix, gap costs settings existence 11 and extension 1, and using conditional compositional score matrix adjustment. Candidate homologues identified from this search included the uncharacterized human protein MB21D2 and the *T. castaneum* sequence XP_969398.1. Pairwise structural comparison between hMB21D2, *Tc*-cGLR and protein structures in the PDB was performed using DALI[32], and Z-scores for homologues less than 90% identical to one another (PDB90) were plotted in GraphPad Prism. A Z-score of 15 for *Tc*-cGLR and 13 for hMB21D2 was selected as a lower cut-off to emphasize directly relevant homologues in analysis.

Following structure determination of hMB21D2 and *T. castaneum* XP_969398.1, predicted cGLRs were further identified in Diptera using PSI-BLAST searches seeded with either *D. melanogaster* CG7194 or the *Tc*-cGLR sequence, selecting in each round for proteins matching known cGLR domain organization and active-site residues. Diptera cGLR sequences were aligned using MAFFT (FFT-NS-i iterative refinement method)[33] and used to construct a phylogenetic tree in Geneious Prime v2020.12.23 using the neighbour-joining method and Jukes–Cantor genetic distance model with no outgroup. Further manual analysis and curation of candidate cGLR sequences were performed based on alignments and predictive structural homology using HHPred[34] and Phyre2[35]. Sequences were selected for predicted structural homology to cGAS, including the presence of a conserved nucleotidyltransferase domain with a G[S/G] activation loop and a [E/D]h[E/D] $X_{50–90}$ [E/D] catalytic triad. Manual refinement was also used to exclude duplicate sequences, gene isoforms and proteins less than 250 or greater than 700 residues. NCBI available genomes from 42 species in Diptera are represented in the final tree, including 31 species in the genus *Drosophila*. Clustering of sequences in the final unrooted tree was used to define clades, with no more than 30% sequence identity shared between members of different clades. Further manual analysis of the tree was used to determine the number and distribution of predicted cGLRs by species (see Extended Data Fig. 2a). PROMALS3D[36] was used for structure guided alignment of apo hcGAS (PDB: 4KM5)[12], hMB21D2 and *Tc*-cGLR in Extended Data Fig. 1a. MAFFT (FFT-NS-i iterative refinement method)[33] was used to align STING sequences in Extended Data Fig. 8a. Geneious Prime software was used to generate the sequence alignments in Fig. 3f and Extended Data Figs. 1a, 3a, 8a.

### Protein expression and purification

Recombinant cGLR and dSTING proteins were expressed and purified using methods previously optimized for hcGAS[14]. Animal cGLR and dSTING sequences were codon-optimized for expression in *Escherichia coli* and cloned from synthetic constructs (GeneArt or Integrated DNA Technologies) into a custom pET16 expression vector with an N-terminal 6× His–MBP fusion tag or an N-terminal 6× His–SUMO2 fusion. The full-length coding sequence was used except for hcGAS 157–522, mouse cGAS 147–607, hMB21D2 S29–F491, *Ds*-cGLR1 19–393 and *D. eugracilis* STING 150–340 as specified. The N terminus of *D. eugracilis* STING 150–340 was fused to the full-length coding sequence of T4 lysozyme connected by a Gly-Ser linker sequence. Briefly, transformed BL21-CodonPlus(DE3)-RIL *E. coli* (Agilent) were grown in MDG media overnight before inoculation of M9ZB media at an $OD_{600}$ of 0.0475. M9ZB cultures were grown to $OD_{600}$ of 2.5 (approximately 5 h at 37 °C with shaking at 230 rpm) followed by cooling on ice for 20 min. Cultures were induced with 500 μM IPTG before incubation

at 16 °C overnight with shaking at 230 rpm. Cultures were pelleted the following day and either flash frozen in liquid nitrogen for storage at −80 °C or directly lysed for purification. Selenomethionine-substituted proteins for crystallography experiments were purified using a modified growth protocol as previously described[37].

For large-scale protein purification, proteins were expressed with a 6× His–SUMO2 (Tc-cGLR, *Ds*-cGLR1, *Deu*-cGLR, *Lc*-cGLR and dSTING) or 6× His–MBP (*Dm*-cGLR1 and *Der*-cGLR1) fusion tag and grown as approximately 4–8 × 1 l cultures in M9ZB media. Pellets were lysed by sonication in lysis buffer (20 mM HEPES pH 7.5, 400 mM NaCl, 30 mM imidazole, 10% glycerol and 1 mM dithiothreitol) and clarified by centrifugation at approximately 47,850*g* for 30 min at 4 °C and subsequent filtration through glass wool. Recombinant protein was purified by gravity flow over Ni-NTA resin (Qiagen). Resin was washed with lysis buffer supplemented to 1 M NaCl and then eluted with 20 ml of lysis buffer supplemented to 300 mM imidazole. SUMO2 fusion proteins were cleaved by supplementing elution fractions with approximately 250 μg of human SENP2 protease (D364–L589 with M497A mutation) during overnight dialysis at 4 °C against dialysis buffer (20 mM HEPES pH 7.5, 250 mM KCl and 1 mM dithiothreitol). MBP-tagged fusion proteins were buffer exchanged into lysis buffer with 4% glycerol and no imidazole to optimize conditions for overnight cleavage by recombinant TEV protease at approximately 10 °C. cGLR proteins were next purified by ion-exchange chromatography using 5 ml HiTrap Heparin HP columns (GE Healthcare) and eluted across a 150–1,000 mM NaCl gradient. Target protein fractions were pooled and further purified by size-exclusion chromatography using a 16/600 Superdex 75 column or 16/600 Superdex 200 column (Cytiva) and storage buffer (20 mM HEPES pH 7.5, 250 mM KCl and 1 mM TCEP). Final proteins were concentrated to approximately 20–30 mg ml⁻¹ and flash frozen with liquid nitrogen and stored at −80 °C for crystallography or supplemented with 10% glycerol before freezing for biochemistry experiments. *Tc*-cGLR and *Ds*-cGLR1 mutant proteins were purified from 1 l M9ZB cultures using Ni-NTA affinity chromatography and overnight dialysis directly into storage buffer (20 mM HEPES pH 7.5, 250 mM KCl, 10% glycerol and 1 mM TCEP) without SUMO2 tag cleavage.

For small-scale protein purification used in the Diptera cGLR screen, recombinant proteins were expressed with a 6× His–MBP fusion tag with the exception of hcGAS, mouse cGAS, *Tc*-cGLR, *Deu*-cGLR, *Lc*-cGLR and *Ds*-cGLR1, which were expressed with a 6× His–SUMO2 fusion tag. Small-scale cultures were grown in 20 ml of M9ZB media, lysed with sonication, and recombinant protein was purified as previously described[9]. Briefly, protein was purified directly from lysates by centrifugation and flow-through over Ni-NTA resin (Qiagen) in 2 ml Mini Spin columns (Epoch Life Sciences). Following elution with elution buffer (20 mM HEPES pH 7.5, 400 mM NaCl, 300 mM imidazole, 10% glycerol and 1 mM dithiothreitol), proteins were buffer exchanged into storage buffer (20 mM HEPES pH 7.5, 250 mM KCl, 10% glycerol and 1 mM TCEP). Fresh protein preparations were immediately used for in vitro nucleotide synthesis reactions.

### Protein crystallization and structure determination

Crystals of native and selenomethionine-substituted hMB21D2 S29–F491, *Tc*-cGLR and T4 lysozyme-dSTING L150–I340 were grown at 18 °C using hanging-drop vapour diffusion. Optimized crystals were grown in EasyXtal 15-well trays (NeXtal Biotechnologies) with 350 μl of reservoir solution and 2-μl drops set with a ratio of 1 μl of protein solution and 1 μl of reservoir solution. hMB21D2 crystals were grown using the reservoir solution (1.2 M ammonium sulfate, 5 mM MgCl$_2$ and 100 mM MES pH 6.2) based on conditions previously identified by Wang and Huang (University of Illinois at Urbana-Champaign)[38] for 1 day before cryoprotection with reservoir solution supplemented with 30% glycerol and freezing in liquid nitrogen. *Tc*-cGLR crystals were grown using the reservoir solution (0.3 M potassium thiocyanate and 10–16% PEG-3350) for 5–16 days before cryoprotection with reservoir

solution supplemented with 15% ethylene glycol and freezing in liquid nitrogen. Apo T4 lysozyme-dSTING crystals were grown using the reservoir solution (0.2 M sodium citrate, 0.1 M Tris-HCl and 22% PEG-3350) 7 days before cryoprotection with reservoir solution supplemented with 15% ethylene and freezing in liquid nitrogen. T4 lysozyme-dSTING–3′2′-cGAMP crystals were grown using the reservoir solution (0.1–0.2 M sodium acetate pH 4.8, 0.2 M ammonium formate and 20–22% PEG-3350) supplemented with 250 μM 3′2′-cGAMP (Biolog) for 10 days before cryoprotection with reservoir solution supplemented to 35% PEG-3350 and freezing in liquid nitrogen. Growth of single hMB21D2 and *Tc*-cGLR crystals was further optimized with streak seeding. X-ray diffraction data were collected at the Advanced Photon Source beamlines 24-ID-C and 24-ID-E and at the Advanced Light Source beamlines 5.0.1 and 8.2.2. Data were processed with XDS and Aimless[39] using the SSRL autoxds script (A. Gonzales, SSRL, Stanford, CA, USA). Experimental phase information for all proteins was determined using data collected from selenomethionine-substituted crystals. Anomalous sites were identified, and an initial map was generated with AutoSol within PHENIX[40]. Structural modelling was completed in Coot[41] and refined with PHENIX. Final structures were refined to stereochemistry statistics for the Ramachandran plot (favoured/allowed), rotamer outliers and MolProbity score as follows: hMB21D2, 97.72%/2.28%, 0.71% and 1.27; *Tc*-cGLR, 98.17%/1.57%, 0.28% and 1.02; dSTING apo, 98.00%/2.00%, 0.33% and 1.30; and dSTING–3′2′-cGAMP, 97.06%/2.86%, 1.72% and 1.63. See Supplementary Table 1 and the 'Data availability' section for deposited PDB codes. All structure figures were generated with PyMOL 2.3.0.

### Nucleotide product synthesis analysis

cGLR nucleotide synthesis activity was analysed by thin-layer chromatography (TLC) as previously described[9]. For the Diptera cGLR screen, recombinant protein preparations were incubated in 10 μl reactions containing 0.5 μl α-[32]P-labelled NTPs (approximately 0.4 μCi each of ATP, CTP, GTP and UTP), 200 μM unlabelled NTPs, 10 mM MgCl$_2$ and 1 mM MnCl$_2$ in a final reaction buffer of 50 mM Tris-HCl pH 7.5, 100 mM KCl and 1 mM TCEP. Reactions were additionally supplemented with approximately 1 μg poly I:C or 5 μM ISD45 dsDNA as indicated. Reactions were incubated at 37 °C overnight and subsequently treated with 1 μl Quick CIP phosphatase (New England Biolabs) for 20 min at 37 °C to remove unreacted phosphate signal. Each reaction was diluted 1:10 in 100 mM sodium acetate pH 5.2, and 0.5 μl was spotted on a 20-cm × 20-cm PEI-cellulose TLC plate. Plates were run with 1.5 M KH$_2$PO$_4$ solvent until approximately 2.5 cm from the top of the plate, dried at room temperature and exposed to a phosphor-screen before signal detection with a Typhoon Trio Variable Mode Imager System (GE Healthcare). For all other nucleotide synthesis reactions visualized by TLC, enzymes were tested at 5 μM with 5 μM nucleic acid ligands and either 1 mM MnCl$_2$ or 10 mM MgCl$_2$ for insect cGLRs or cGAS, respectively. hMB21D2 activity was tested with 1 mM MnCl$_2$ and 10 mM MgCl$_2$ using the following synthetic innate immune agonists: lipopeptide Pam3CSK4 (Invivogen), *Staphylococcus aureus* lipoteichoic acid (LTA-SA; Invivogen), *Saccharomyces cerevisiae* cell wall preparation (Zymosan; Invivogen), *Bacillus subtilis* peptidoglycan (PGN-BS; Invivogen), synthetic lipid A mimic (CRX-527; Invivogen), *B. subtilis* flagellin (FLA-BS; Invivogen), imidazoquinoline (Imiquimod; Invivogen), CpG oligonucleotide (ODN 2006; Invivogen) and *S. aureus* 23S rRNA oligonucleotide (ORN Sa19; Invivogen). Other than Diptera screen reactions, samples were not diluted in sodium acetate before spotting on PEI-cellulose TLC plates. TLC images were adjusted for contrast using FIJI[42] and quantified using ImageQuant (8.2.0). Nucleotide product formation was measured according to the ratio of product to total signal for each reaction. For Figs. 1c, 2d and Extended Data Figs. 3c, d, 4b, 5c, relative activity was calculated as the percent conversion for each reaction relative to maximal conversion observed by wild-type enzyme or in the presence of 40-bp dsRNA for insect cGLRs and 45-bp dsDNA for cGAS.

### Electrophoretic mobility shift assay

Analysis of in vitro protein–nucleic acid complex formation was conducted as previously described[14]. Briefly, 1 μM 40-bp dsRNA or 45-bp dsDNA was incubated with *Ds*-cGLR1 or hcGAS NTase domain (D157–522) at a concentration of 0.5, 1 or 2 μM. Complex formation was performed with the final reaction buffer (20 mM HEPES-NaOH pH 7.8, 75 mM KCl and 1 mM dithiothreitol. Reactions (20 μl) were incubated at 4 °C for 20 min before separation on a 2% agarose gel using 0.5× TB buffer (45 mM Tris and 45 mM boric acid) as a running buffer. The agarose gel was post-stained in 0.5× TB buffer supplemented with 10 μg ml⁻¹ ethidium bromide with gentle shaking at 25 °C for 45 min. Complex formation was visualized using a ChemiDoc MP Imaging System (Bio-Rad).

### In vitro condensate formation assays

In vitro condensate formation was analysed as previously described with minor modifications[18,43]. Briefly, *Ds*-cGLR1 was labelled with AlexaFluor-488 (AF488) carboxylic acid (succinimidyl ester) (Thermo Fisher Scientific) according to the manufacturer's manuals using a molar ratio of 1:10 at 4 °C for 4 h. Excess free dye was removed by dialysis against buffer (20 mM HEPES-KOH pH 7.5, 250 mM KCl and 1 mM dithiothreitol) at 4 °C overnight, and AF488-labelled *Ds*-cGLR1 was then further purified on a PD-10 desalting column (GE Healthcare) eluted with storage buffer (20 mM HEPES-KOH pH 7.5, 250 mM KCl and 1 mM TCEP). Final AF488-labelled *Ds*-cGLR1 was concentrated to approximately 5 mg ml⁻¹, flash frozen in liquid nitrogen and stored as aliquots at −80 °C. hcGAS and hcGAS NTase domain (D157–F522) proteins were prepared as previously described[43].

To induce condensate formation, *Ds*-cGLR1 (10 μM, containing 1 μM AF488-labelled *Ds*-cGLR1) was mixed with various lengths of RNA (10 μM each) in buffer (20 mM Tris-HCl pH 7.5, 1 mg ml⁻¹ BSA and 1 mM TCEP) in the presence of various salt concentrations at 25 °C in a total reaction volume of 20 μl. The details of proteins, nucleic acids and salt concentrations are indicated in the figures. *Ds*-cGLR1–RNA reactions were placed in 384-well non-binding microplates (Greiner Bio-One) and incubated at 25 °C for 30 min before imaging to allow condensates to settle. Fluorescence microscopy images were acquired at 25 °C using a Leica TCS SP5 X (Leica Microsystems) mounted on an inverted microscope (DMI6000; Leica Microsystems) with an oil immersion ×63/numerical aperture 1.4 objective lens (HCX PL APO; Leica Microsystems). AF488-labelled *Ds*-cGLR1, hcGAS and hcGAS NTase domain proteins were detected with excitation at 488 nm (emission at 500–530 nm). Microscopy images were processed with FIJI[42] and contrast adjusted with a uniform threshold setup for each enzyme.

### Cellular STING signalling assays

Human HEK293T cells were purchased directly from the American Type Culture Collection (ATCC) and were maintained in complete media (DMEM supplemented with penicillin, streptomycin and 10% FBS) at 37 °C. HEK293T cells were validated by the ATCC and were not tested for mycoplasma contamination. For all assays, 4.5 × 10⁴ cells were plated in 96-well plates. STING and cGLR activity assays were performed using the Dual-Luciferase Reporter Assay System (Promega) as previously described[12], with modifications. Lipofectamine-2000 was used to transfect IFNβ-firefly luciferase and TK-Renilla luciferase reporters and 5 ng of pcDNA4–mouse STING or 15 ng of pcDNA4–dSTING hybrid construct (human STING transmembrane domains fused to the *D. eugracilis* STING CDN-binding domain (L150–I340) appended with the human STING C-terminal tail). For cGLR signalling assays, 150 ng of *Drosophila* cGLR1, 30 ng hcGAS with 120 ng empty vector, or 150 ng empty vector were additionally transfected. The native coding sequence was used for each cGLR and STING pcDNA4 plasmid. Twenty-four to thirty hours after transfection, luciferase was measured using a GloMax microplate reader (Promega), and relative IFNβ expression was calculated

by normalizing firefly to Renilla readings. For poly I:C stimulation of cGLR activity, cells were transfected with 100 ng poly I:C (6.125–200 ng for titration experiment) 5 h after plasmid transfection. For dSTING signalling assays, a final concentration of 500 pM to 50 μM 2′3′-cGAMP or 3′2′-cGAMP was delivered to cells using a digitonin permeabilization buffer[44] 10 h before luciferase measurement.

## Nucleotide purification and HPLC analysis

Enzymatic synthesis of cGLR nucleotide products for HPLC analysis was performed using 100-μl reactions containing 10 μM cGLR enzyme, 200 μM ATP, 200 μM GTP, 10 μg poly I:C, 1 mM $MnCl_2$ and 50 mM Tris-HCl pH 7.5. Protein storage buffer (20 mM HEPES pH 7.5, 250 mM KCl and 1 mM TCEP) was used as necessary to adjust KCl concentration to approximately 100 mM. Reactions were incubated at 37 °C for 1 h and then nucleotide product was recovered by filtering reactions through a 30-kDa cut-off concentrator (Amicon) to remove protein. Nucleotide products were separated on an Agilent 1200 Infinity Series LC system using a C18 column (Zorbax Bonus-RP 4.6 × 150 mm, 3.5 μm) at 40 °C. Products were eluted at a flow rate of 1 ml min$^{-1}$ with a buffer of 50 mM $NaH_2PO_4$ pH 6.8 supplemented with 3% acetonitrile.

To purify the *Deu*-cGLR product for mass spectrometry analysis, nucleotide synthesis reaction conditions were scaled as previously described for bacterial cGAS/DncV-like nucleotidyltransferase reactions[9,45]. Briefly, a 10-ml reaction containing 528 nM *Deu*-cGLR enzyme, 125 μM ATP, 125 μM GTP, approximately 250 μg poly I:C, 1 mM $MnCl_2$, 50 mM Tris-HCl 7.5 and approximately 25 mM KCl was incubated with gentle rotation for 36 h at 37 °C follow by Quick CIP (NEB) treatment for 6 h. The reaction was monitored using a 20 μl aliquot supplemented with α-$^{32}$P-labelled NTPs and to visualize product formation by TLC. Following incubation, the large-scale reaction was filtered through a 10-kDa concentrator (Amicon) and purified by anion-exchange chromatography using a 1-ml Q-sepharose column (Cytiva) washed with water and eluting with a 0–2 M ammonium acetate gradient. Fractions corresponding to the main product 3′2′-cGAMP were differentiated from fractions corresponding to 2′3′-c-di-AMP by HPLC analysis. Product fractions were further purified by size-exclusion chromatography using a Superdex 30 Increase 10/300 GL (Cytiva) with $dH_2O$ as a running buffer. Peak fractions were eluted in 1-ml volumes, pooled and evaporated for storage before mass spectrometry analysis.

## Nucleotide mass spectrometry analysis and 3′2′-cGAMP identification

Purified nucleotide product samples were evaporated at 40 °C under a gentle nitrogen stream. The residual pellet was resuspended in 200 μl HPLC grade water (J.T. Baker), and 40 μl was then mixed with 40 μl of water containing 50 ng ml$^{-1}$ tenofovir as internal standard and transferred to measuring vials.

Experiments for 3′2′-cGAMP identification were performed on an ACQUITY UPLC I-Class/Vion IMS-QTOF high-resolution LC−MS system (Waters Corporation). Reverse-phase chromatographic separation was carried out at 30 °C on a C18 column (Nucleodur Pyramid C18 50 × 3 mm; 3 μm Macherey Nagel) connected to a C18 security guard (Phenomenex) and a 2-μm column saver. Separation was achieved using a binary gradient of water containing 10 mM ammonium acetate and 0.1% acetic acid (solvent A) and methanol (solvent B). The analytes were eluted at a flow rate of 0.6 ml min$^{-1}$. The eluting programme was as follows: 0–4 min: 0% B, 4–7.3 min: 0–10% B. This composition of 10% B was held for 1 min, then the organic content was increased to 30% within 2.7 min. The column was then re-equilibrated to 0% B for 2 min. The total analysis run time was 13 min. High-resolution mass spectrometry data were collected on a Vion IMS-QTOF mass spectrometer equipped with an electrospray ionization source, operating in positive ionization mode. The capillary voltage was set at 2.5 kV and the cone voltage at 40 V. The source temperature and desolvation gas temperature was 150 °C and 600 °C, respectively. Analyte

fragmentation was achieved using argon as the collision gas. Collision energy of 10 V was used to obtain a low collision energy spectrum. For high collision energy spectrum, the collision energy was ramped from 15 to 30 V. Data acquisition was controlled by the UNIFI 1.9.4.0 software (Waters). For 3′2′-cGAMP identification, the retention times, drift times and fragment spectra of a synthetic 3′2′-cGAMP standard (Biolog) were collected as a reference and compared with those of the suspected 3′2′-cGAMP in the samples.

## 3′2′-cGAMP quantification

For quantification of 3′2′-cGAMP, chromatographic conditions were transferred to a API4000 mass spectrometer (Sciex) coupled to a Shimadzu HPLC system (Shimadzu). The analytes were ionized by means of electrospray ionization in positive mode applying an ion spray voltage of 3,000 V. Further electrospray ionization parameters were as follows: curtain gas (CUR): 30 psi; collision gas (CAD): 9; source temperature: 650 °C; gas 1: 60 psi and gas 2: 45 psi, respectively. Detection was performed in SRM mode, selecting first for the double-protonated parent ion of 3′2′-cGAMP and 3′3′-cGAMP (used in calibrator series). This resulted in the following mass transitions: 3′2′-cGAMP and 3′3′-cGAMP: $m/z$ 338.2 → 152 (quantifier), $m/z$ 338.2 → 136 (identifier). Tenofovir served as the internal standard ($m/z$ 288 → 176).

For 3′2′-cGAMP semiquantitative quantification from lysate samples in the Diptera cGLR screen, calibration curves were created by plotting peak area ratios of 3′3′-cGAMP as an internal standard versus the nominal concentration of the calibrators. The calibration curve was calculated using quadratic regression and 1/x weighting.

## Synthetic cyclic dinucleotide standards

Synthetic nucleotide standards used for HPLC analysis and mass spectrometry analysis were purchased from Biolog Life Science Institute: 3′3′-cGAMP (cat no. C 117), 2′3′-cGAMP (cat no. C 161), 3′2′-cGAMP (cat no. C 238), 2′3′-c-di-AMP (cat no. C 187) and 2′3′-c-di-GMP (cat no. C 182).

## Nuclease P1 and poxin cleavage analysis

Nuclease P1 cleavage analysis was performed using *Dm*-cGLR1 reactions labelled with either α-$^{32}$P-ATP or α-$^{32}$P-GTP as previously described[9,19]. Briefly, radiolabelled nucleotide products were incubated with nuclease P1 (80 mU; N8630, Sigma) in buffer (30 mM NaOAc pH 5.3, 5 mM $ZnSO_4$ and 50 mM NaCl) for 30 min in the presence of Quick CIP (NEB).

Poxin cleavage reactions were carried out using purified insect viral AcNPV enzyme as previously described[22,37]. For HPLC analysis of poxin cleavage, 100-μl reactions were performed using 100 μM synthetic 2′3′-cGAMP or 3′2′-cGAMP, 50 nM AcNPV poxin, 50 mM HEPES pH 7.5, 10 mM KCl and 1 mM TCEP. Reactions were incubated at 37 °C and at each specified time reactions were terminated by heat inactivation at 95 °C for 2 min before HPLC analysis as described above. For TLC analysis of poxin cleavage, reactions were performed using α-$^{32}$P-GTP-labelled 2′3′-cGAMP synthesized by mcGAS or 3′2′-cGAMP synthesized by *Deu*-cGLR in 5-μl reactions containing 2.5 μM nucleotide product and 1 μM AcNPV poxin, 50 mM HEPES pH 7.5, 10 mM KCl and 1 mM TCEP. Reactions were incubated at 37 °C and at each specified time reactions were terminated by heat inactivation at 80 °C for 5 min before PEI-cellulose TLC analysis as described above.

## STING CDN thermal shift assay

A final concentration of 15 μM dSTING was mixed with 3× SYPRO orange dye and 100 μM synthetic CDN (Biolog) (or a 40 nM to 100 μM concentration gradient as described in Extended Data Fig. 8c) in 20 mM HEPES-KOH pH 7.5 and 100 mM KCl. Samples were heated from 20 to 95 °C in a Bio-Rad CFX thermocycler with HEX channel fluorescence measurements taken every 0.5 °C. The derivative of each curve over time was calculated using GraphPad Prism and graphed as a percent maximum change in fluorescence or used to calculate the melting temperature.

## D. melanogaster cyclic dinucleotide injection and signalling analysis

Fly stocks were raised on standard cornmeal agar medium at 25 °C. All fly lines used in this study were *Wolbachia* free. $w^{1118}$, *dSTING*$^{Control}$ and *dSTING*$^{Rxn}$ stocks have been described previously[23,26]. *Relish*$^{E20}$ flies isogenized to the DrosDel $w^{1118}$ isogenic background were a kind gift from L. Teixeira (Instituto Gulbenkian de Ciência)[46]. Cyclic dinucleotides including 3′2′-cGAMP (Biolog), 2′,3′-cGAMP (Invivogen) and 3′3′-c-di-GMP (Invivogen) were dissolved in 10 mM Tris-HCl pH 7.5 and diluted to the indicated concentrations. Adult flies (3–5-day old) were injected with 69 nl of cyclic dinucleotide solution or 10 mM Tris-HCl pH 7.5 (negative control) by intrathoracic injection using a Nanoject II apparatus (Drummond Scientific). Flies were collected 24 h later in pools of 6 individuals (3 males and 3 females) or 10 individuals (5 males and 5 females) and homogenized for RNA extraction and quantitative PCR with reverse transcription (RT–qPCR) analysis, as described[26]. The sample size for all *Drosophila* experiments was determined using previously published protocols[26]. Flies were randomly selected for each experimental group and blinding was not performed.

## D. melanogaster viral challenge assays

For 3′2′-cGAMP and virus co-injection, flies were injected with 69 nl of virus (DCV: 5 plaque-forming units (p.f.u.), vesicular stomatitis virus (VSV): 2,000 p.f.u.) in 10 mM Tris-HCl pH 7.5 or in a 0.9 mg ml$^{-1}$ 3′2′-cGAMP solution. For titration experiments comparing cGAMP isomers, 69 nl of DCV (5 p.f.u.) in serial diluted concentrations of 2′3′-cGAMP or 3′2′-cGAMP were injected in the body cavity of the flies. Survival was monitored daily, and flies were collected in pools of 6 individuals (3 males and 3 females) or 10 individuals (5 males and 5 females) at the indicated time points to monitor the viral RNA load by RT–qPCR.

## Statistical analyses

All statistical analyses were performed using GraphPad Prism 9.0.1. Error bars and sample size for each experiment are defined in the figure legends. Comparisons between groups for gene expression and viral loads were analysed by unpaired parametric $t$-test, two-tailed with no corrections; comparison between groups for survival curves following viral infection were analysed by log-rank test.

## Reporting summary

Further information on research design is available in the Nature Research Reporting Summary linked to this paper.

## Data availability

Coordinates and structure factors of hMB21D2, *Tc*-cGLR, dSTING and the dSTING–3′2′-cGAMP complex have been deposited in the PDB under the accession codes 7LT1, 7LT2, 7MWY and 7MWZ. All other data are available in the paper or the supplementary materials. Source data are provided with this paper.

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

**Acknowledgements** We are grateful to P. M. Devant, L. Liu, K. Chat, A. Holleufer, R. Hartmann, T. I. V. Ly, and members of the Kranzusch laboratory for helpful comments and discussion; C. de Oliveira Mann for assistance developing the mass spectrometry analysis of insect cGLRs; and M. Burroughs and A. Iyer for assistance with the bioinformatics analysis of cGAS-like enzymes. The work was funded by grants to P.J.K. from the Pew Biomedical Scholars program, the Burroughs Wellcome Fund PATH program, The Richard and Susan Smith Family Foundation, The Mathers Foundation, The Mark Foundation for Cancer Research, a Cancer Research Institute CLIP grant, a V Foundation V Scholar Award, and the Parker Institute for Cancer Immunotherapy; grants to J.-L.I. and H.C. from the Agence Nationale de la Recherche (ANR-17-CE15-0014), the Investissement d'Avenir Programme (ANR-10-LABX-0036 and ANR-11-EQPX-0022), the Institut Universitaire de France, the Chinese National Overseas Expertise Introduction Center for Discipline Innovation (Project '111' (D18010)), the Foreign Experts Program (2020A1414010306) and The Natural Science Foundation (32000662); a grant from the Deutsche Forschungsgemeinschaft (DFG) within the Priority Program SPP1879 and INST 192/524-1 FUGG; and a grant to A.S.Y.L. from the Pew Biomedical Scholars program. W.Z. is supported as a Benacerraf Fellow in Immunology and through a Charles A. King Trust Postdoctoral Fellowship. B.R.M. is supported as a Ruth L. Kirschstein NRSA Postdoctoral Fellow NIH F32GM133063. X-ray data were collected at the Northeastern Collaborative Access Team beamlines 24-ID-C and 24-ID-E (P30 GM124165), and used a Pilatus detector (S10RR029205), an Eiger detector (S10OD021527) and the Argonne National Laboratory Advanced Photon Source (DE-AC02-06CH11357), and at beamlines 5.0.1 and 8.2.2 of the Advanced Light Source, a US DOE Office of Science User Facility under contract no. DE-AC02-05CH11231 and supported in part by the ALS-ENABLE program and the NIGMS grant P30 GM124169-01.

**Author contributions** Experiments were designed and conceived by K.M.S. and P.J.K. Gene identification and phylogenetic analyses were performed by K.M.S., B.R.M. and P.J.K. *Tc*-cGLR structural experiments were performed by K.M.S. hMB21D2 structural experiments were performed by B.R.M. dSTING structural experiments were performed by K.M.S. and A.E.R. hMB21D2, cGLR and dSTING biochemical experiments were performed by K.M.S. and A.E.R. Phase separation and RNA-binding analysis were performed by W.Z. and A.E.R. Cell biology experiments were designed by K.M.S. and A.S.Y.L., and performed by K.M.S. Nucleotide purification and mass spectrometry experiments were performed by K.M.S., H.B., M.K. and R.S. In vivo *Drosophila* experiments were designed and performed by X.A., Y.C., L.L., Z.W., H.C. and J.-L.I. The manuscript was written by K.M.S. and P.J.K. All authors contributed to editing the manuscript and support the conclusions.

**Competing interests** The authors declare no competing interests.

**Additional information**
**Correspondence and requests for materials** should be addressed to P.J.K.

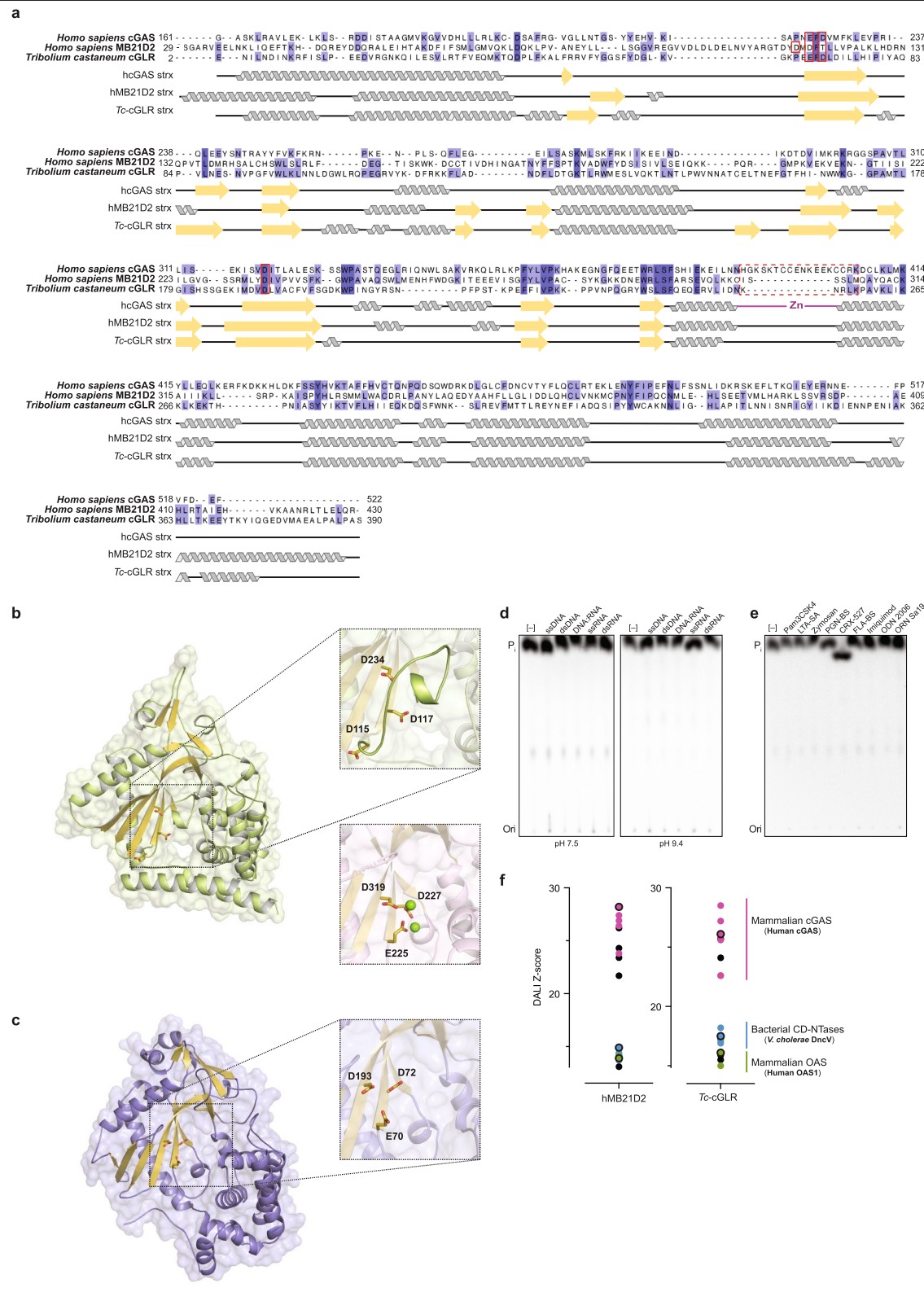

**Extended Data Fig. 1** | See next page for caption.

**Extended Data Fig. 1 | Sequence and structural analysis of hMB21D2 and *Tc*-cGLR. a**, Structure guided sequence alignment of the catalytic domain of hcGAS (PDB: 4KM5)[12], hMB21D2 and *Tc*-cGLR. Strict secondary structure conservation further supports conserved structural homology despite primary sequence divergence. The [D/E]hD[X$_{50-90}$]D catalytic triad is highlighted with a red outline and the human Zn-ribbon insertion that is absent in other cGLRs is denoted with a red dashed outline. hMB21D2 contains an additional 61 residues that are not resolved in the crystal structure and are absent from the alignment. **b**, **c**, Zoomed-in cutaways of the hMB21D2 (**b**) and *Tc*-cGLR (**c**) crystal structures highlighting positioning of conserved catalytic residues in the nucleotidyltransferase active site. In hcGAS, the analogous residues coordinate two Mg$^{2+}$ metal ions to control synthesis of 2′3′-cGAMP (inset, middle; PDB: 6CTA)[14]. The hMB21D2 structure is in an inactive state distinguished by misaligned catalytic residues and occlusion of the active site by an extended Gly-Gly activation loop, indicating that catalytic activation is probably controlled by a conformational rearrangement. **d**, **e**, TLC analysis of in vitro tests for potential activating ligands of hMB21D2. No nucleotide products were identified upon stimulation with 40-nt or 40-bp nucleic acid ligands (**d**) or ligands known to activate mammalian Toll-like receptors (**e**). Data shown are representative of $n = 3$ independent experiments. **f**, Z-score structural similarity plot showing homology between hMB21D2 and *Tc*-cGLR with representative structures in the PDB (PDB90). Increasing Z-score indicates greater homology, confirming the close relationship between animal cGLR enzymes and mammalian cGAS, and more distant similarity to cGAS/DncV-like nucleotidyltransferases (CD-NTases) in bacterial antiphage defence systems and human oligoadenylate synthase 1 (refs. [9,15,45,47–50]). Z-score cut-offs are 13 and 15 for hMB21D2 and *Tc*-cGLR, respectively.

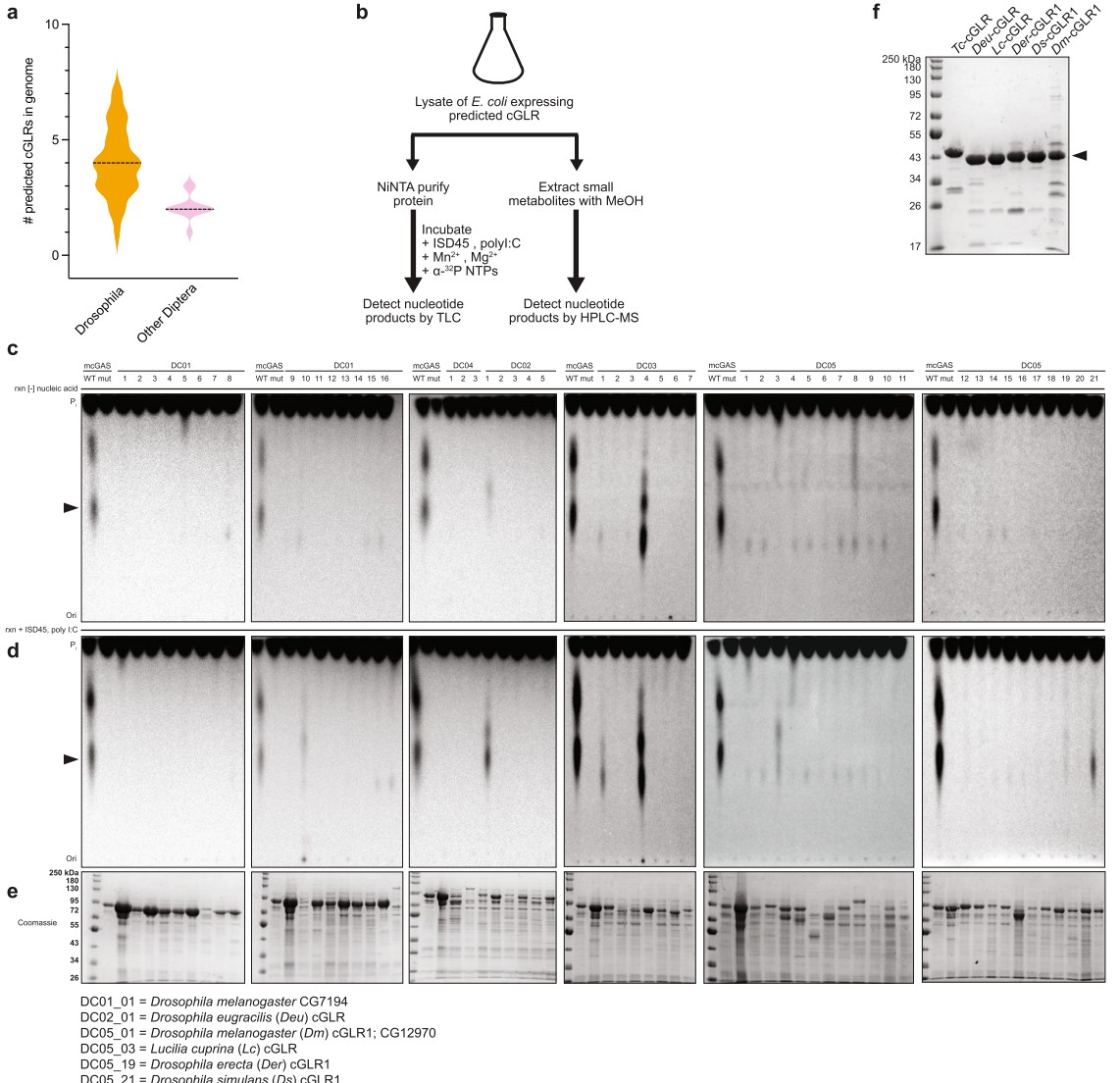

DC01_01 = *Drosophila melanogaster* CG7194
DC02_01 = *Drosophila eugracilis* (*Deu*) cGLR
DC05_01 = *Drosophila melanogaster* (*Dm*) cGLR1; CG12970
DC05_03 = *Lucilia cuprina* (*Lc*) cGLR
DC05_19 = *Drosophila erecta* (*Der*) cGLR1
DC05_21 = *Drosophila simulans* (*Ds*) cGLR1

**Extended Data Fig. 2 | Forward biochemical screen of predicted cGLRs in Diptera. a**, Violin plot showing the number of predicted cGLRs in Diptera genomes. *Drosophila* genomes (*n* = 31 species) have a median of four predicted cGLRs in contrast to a median of two predicted cGLRs in other dipteran insects (*n* = 11 species). **b**, Schematic of the in vitro screen of predicted cGLRs in the order Diptera. Fifty-three sequences were selected representing each clade in the phylogeny in Fig. 2a. Following recombinant protein expression in *E. coli*, lysates were split into two samples for parallel TLC analysis of in vitro enzymatic activity and HPLC-MS analysis of lysate nucleotide metabolites. **c**, **d**, Purified cGLR proteins were incubated overnight at 37 °C with α$^{32}$P-radiolabelled nucleotides, a mixture of Mn$^{2+}$ and Mg$^{2+}$, and the 45-bp immunostimulatory DNA ISD45 or the synthetic dsRNA analogue poly I:C as potential nucleic acid ligands, and reactions were visualized by PEI-cellulose

TLC. Wild-type (WT) and catalytically inactive mouse cGAS enzymes were used as controls for each sample set. Note that mouse cGAS exhibits dsDNA-independent activity in the presence of Mn$^{2+}$ (ref. [51]). Predicted Diptera cGLRs are grouped by clade (DC01–05) and numbered within each clade. Ligand-dependent activity was identified for DC02_01, DC05_03, DC05_19 and DC05_21; species listed below. We observed ligand-independent activity for two enzymes in clade 3. Data represent *n* = 2 independent experiments. **e**, SDS–PAGE and Coomassie stain analysis of NiNTA-purified cGLR protein fractions used for the biochemical screen. **f**, SDS–PAGE and Coomassie stain analysis of final cGLR proteins used for biochemical studies, which were purified by NiNTA-affinity, ion-exchange chromatography and size-exclusion chromatography.

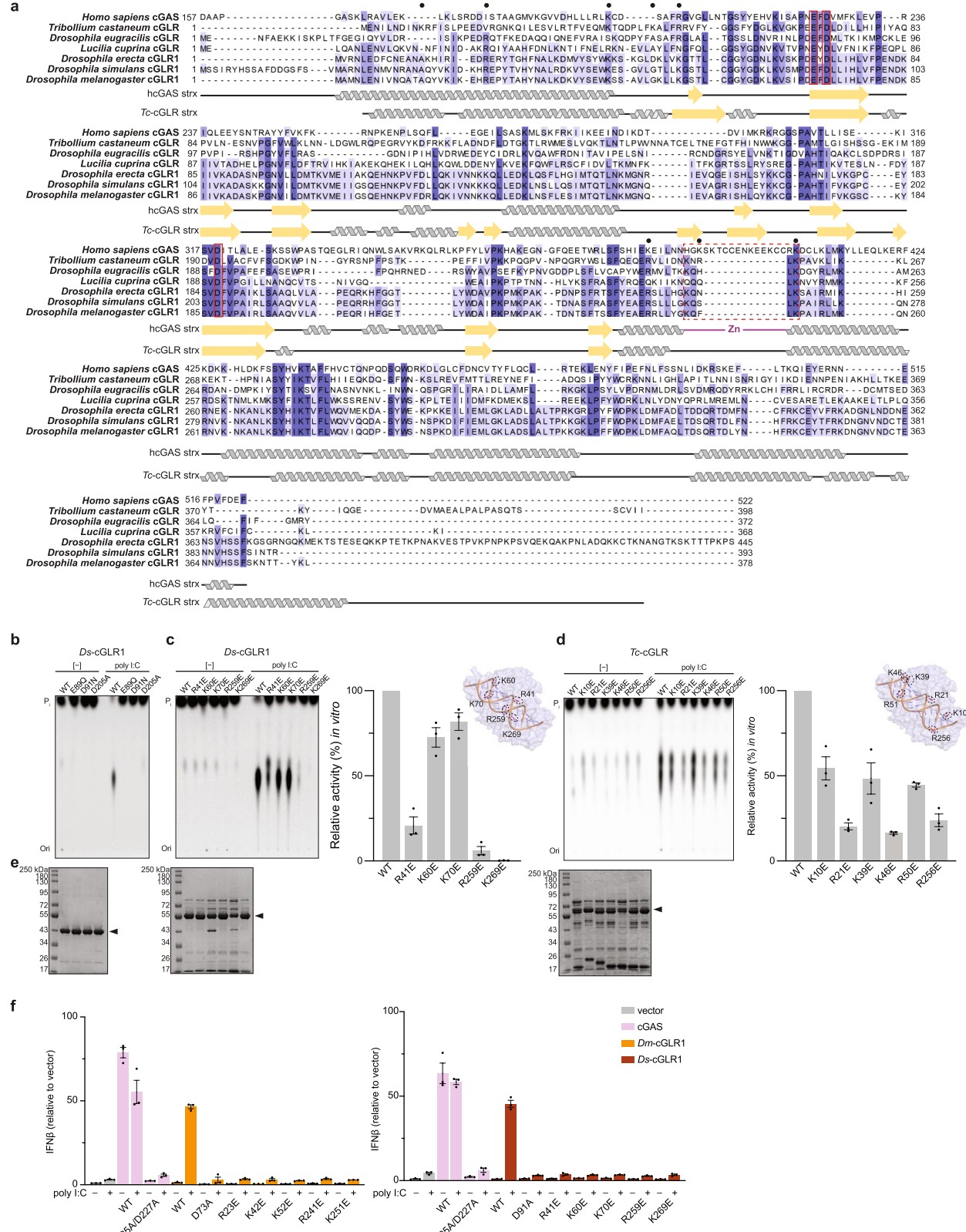

**a**

**b**

**c**

**d**

**e**

**f**

**Extended Data Fig. 3** | See next page for caption.

**Extended Data Fig. 3 | Sequence analysis and mutagenesis of insect cGLRs.**
**a**, Alignment of the catalytic domain of hcGAS and active cGLRs identified in
*T. castaneum*, *D. eugracilis*, *L. cuprina*, *D. erecta*, *D. simulans* and *D. melanogaster*.
The EhD[X$_{50-90}$]D catalytic triad is highlighted with a red outline and the human
Zn-ribbon insertion that is absent in insect cGLRs is denoted with a red dashed
outline. Predicted basic ligand-binding residues selected for mutational
analysis denoted by black circles. cGLRs from *D. erecta* and *D. simulans* are
close homologues of *Dm*-cGLR1 (76% and 91% sequence identity, respectively)
and thus are also referred to as 'cGLR1'. All biochemical experiments with *Ds*-
cGLR1 were performed with a construct beginning at M19. **b**, In vitro reactions
demonstrating that mutation of the catalytic residues ablates nucleotide
product synthesis by *Ds*-cGLR1 in response to poly I:C. **c**, **d**, In vitro reactions
analysing dsRNA recognition through the putative ligand-binding surface by
*Ds*-cGLR1 (**c**) or *Tc*-cGLR (**d**). The insets for panels **c** and **d** show models of the
*Tc*-cGLR–dsRNA complex based on the hcGAS–dsDNA structure (PDB: 6CTA)[14],
indicating predicted dsRNA-interacting residues in *Ds*-cGLR1 (**c**) or *Tc*-cGLR
(**d**). Charge swap mutation to these residues variably disrupted poly I:C-
stimulated activity by *Ds*-cGLR1 and *Tc*-cGLR, shown by TLC (left) and
quantified relative to WT activity (right). Data in **b**–**d** are representative of $n = 3$
independent experiments. **e**, SDS–PAGE and Coomassie stain analysis of
purified WT and mutant proteins, as labelled in the above TLC images. **f**, IFNβ
luciferase assay in which cGLRs are expressed in human cells and CDN synthesis
is detected by mammalian STING activation, as in Fig. 2e. IFNβ was quantified
relative to the empty vector control. In comparison to hcGAS control, which is
activated by expression vector-plasmid DNA, *Dm*-cGLR1 (left) and *Ds*-cGLR1
(right) strictly require poly I:C stimulation to activate a downstream STING
response. Mutation to catalytic residues or putative ligand-binding residues
ablates cGLR1 signalling. See Fig. 2e: *Dm*-cGLR1 activity quantified relative to
WT activity upon poly I:C stimulation. Data are mean ± s.e.m. of $n = 3$ technical
replicates and representative of $n = 3$ independent experiments.

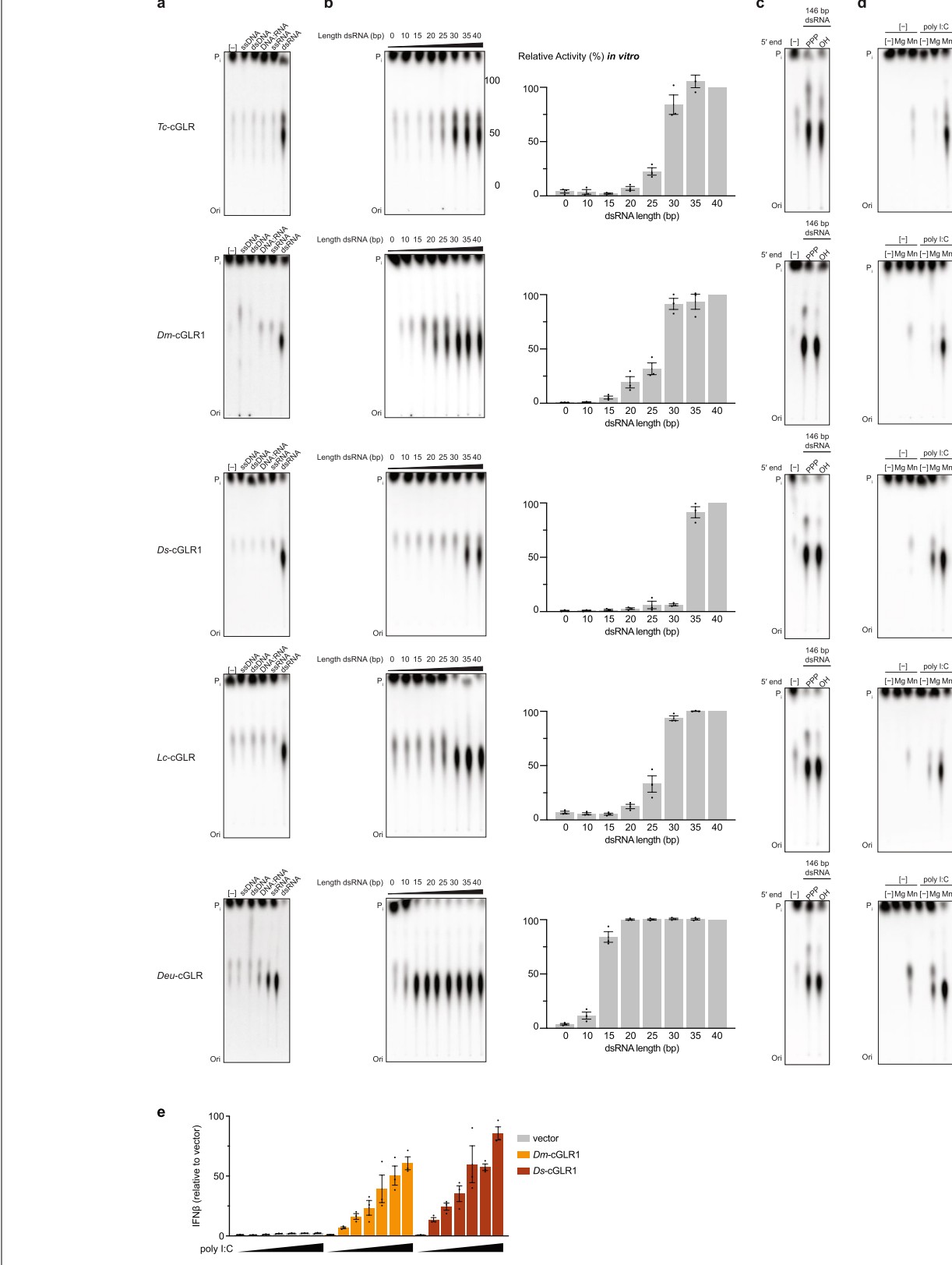

**Extended Data Fig. 4** | See next page for caption.

**Extended Data Fig. 4 | Analysis of RNA recognition by insect cGLRs.**
**a**–**c**, In vitro activity assays for each active insect cGLR demonstrating that dsRNA recognition is required for enzyme activation. Reactions were performed with 40-nt or 40-bp synthetic ligands. Weak *Deu*-cGLR ssRNA-stimulated activity may be explained by transient short duplex formation similar to observations that some ssDNA oligos can stimulate mouse cGAS dsDNA-dependent activity[3]. **b**, TLC and quantification for enzyme activation in the presence of a panel of 10–40-bp synthetic dsRNA ligands. dsRNA (30 bp) is sufficient to stimulate maximal activity for *Tc*-cGLRs, *Dm*-cGLRs and *Lc*-cGLRs, while *Ds*-cGLR1 requires 35 bp and *Deu*-cGLR can be activated by dsRNAs as short as 15 bp. Data are mean ± s.e.m., quantified relative to maximum observed activity. **c**, Reactions with 146-bp in vitro-transcribed dsRNAs containing either a 5′ triphosphate or 5′ OH termini demonstrate that dsRNA recognition by insect cGLRs does not involve 5′-end discrimination. **d**, Deconvolution of catalytic metal requirements for enzymatic activity by insect cGLRs. Insect cGLRs require $Mn^{2+}$ for maximal catalytic activity, with weak product formation observed in the presence of $Mg^{2+}$. **e**, Poly I:C titration demonstrates that dsRNA stimulation of *Drosophila* cGLR1 activity in cells is dependent on RNA concentration. IFNβ luciferase assay in which cGLRs are expressed in human cells and CDN synthesis is measured by mammalian STING activation, as in Fig. 2e and Extended Data Fig. 3f. IFNβ quantified relative to the empty vector control. Data are mean ± s.e.m. of $n = 3$ technical replicates. All data in **a**–**e** represent $n = 3$ independent experiments.

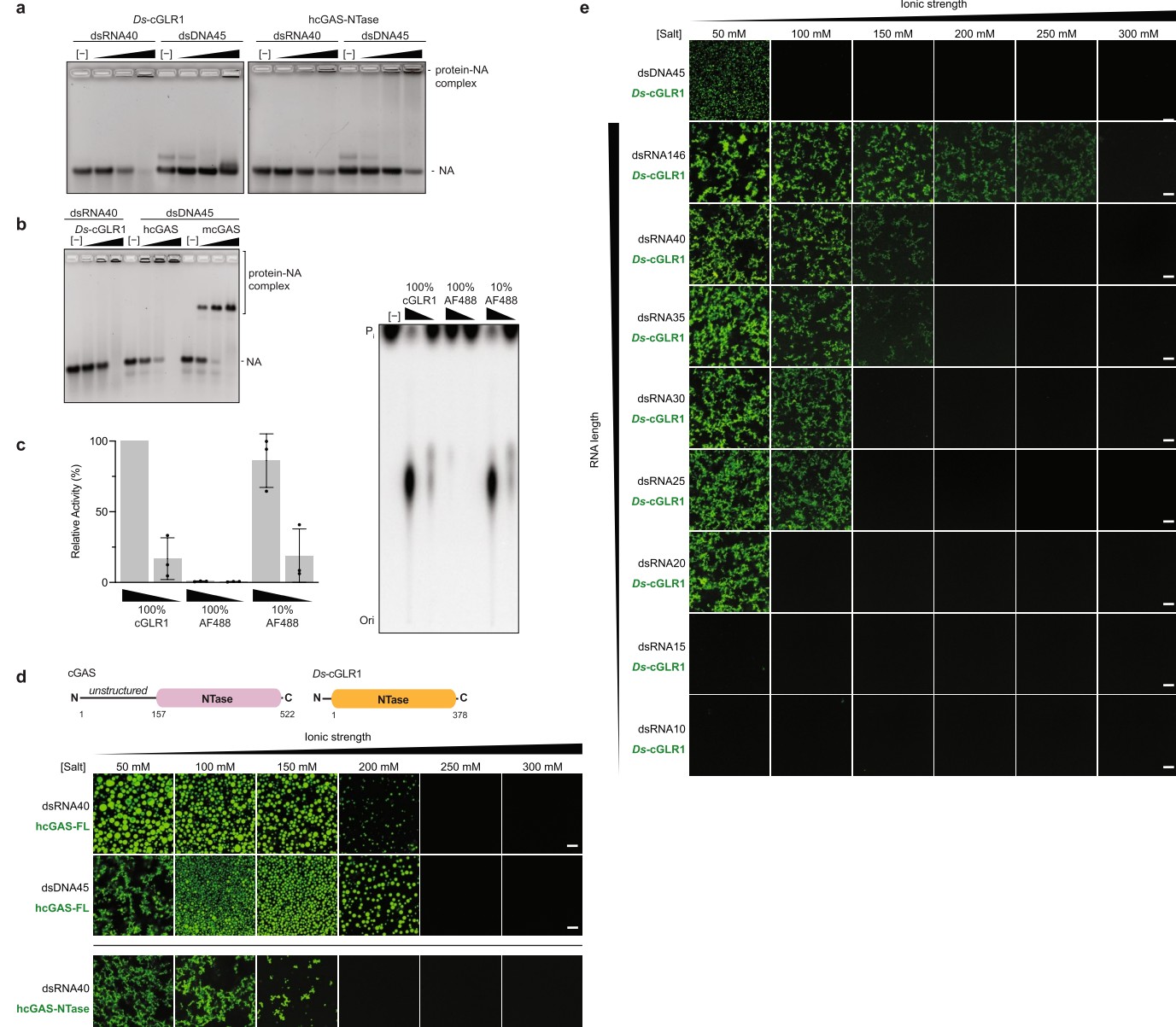

**Extended Data Fig. 5 | Characterization of *Ds*-cGLR1–dsRNA condensate formation. a**, Electrophoretic mobility shift assay (EMSA) showing binding between *Ds*-cGLR1 or the C-terminal NTase domain of hcGAS (hcGAS-NTase) and a 40-bp dsRNA or 45-bp dsDNA. *Ds*-cGLR1 preferentially binds to dsRNA and more weakly interacts with dsDNA, consistent with observed binding between hcGAS and dsRNA[11]. **b**, EMSA comparison of *Ds*-cGLR1–dsRNA binding and mammalian cGAS–dsDNA binding. Similar to hcGAS, *Ds*-cGLR1 forms a higher-order protein–nucleic acid complex that does not migrate through the gel, in contrast to the 2:2 binding observed between mouse cGAS and dsDNA. Data in **a** and **b** are representative of *n* = 3 independent experiments. **c**, Analysis of the effect of AF488 labelling on *Ds*-cGLR1 enzymatic activity. Similar to previous observations with hcGAS[43], AF488 labelling negatively impacts enzymatic activity but has minimal effect at the ratio of 90% unlabelled and 10% labelled protein used for all imaging experiments. Data are mean ± s.e.m. of

*n* = 3 independent experiments. **d**, **e**, Analysis of hcGAS (**d**) and *Ds*-cGLR1 (**e**) phase separation with AF488-labelled protein. Mammalian cGAS contains a highly disordered N-terminal extension of approximately 150 residues, but this unstructured extension is absent in insect cGLR sequences. In the presence of dsDNA, full-length hcGAS forms highly dynamic liquid droplets[18,43,52], whereas the minimal hcGAS NTase domain forms rigid protein–DNA condensates similar to those formed by *Ds*-cGLR1–RNA complexes. hcGAS exhibits a preference for condensate formation in the presence of dsDNA (**d**), whereas *Ds*-cGLR1 exhibits a preference for dsRNA (**e**), as observed in panel **a**. Scale bars, 10 μm. Analysis of *Ds*-cGLR1 dsRNA length specificity for condensate formation demonstrates clear length dependency (**e**) and supports that long dsRNA and condensate formation are required for maximal *Ds*-cGLR1 activation.

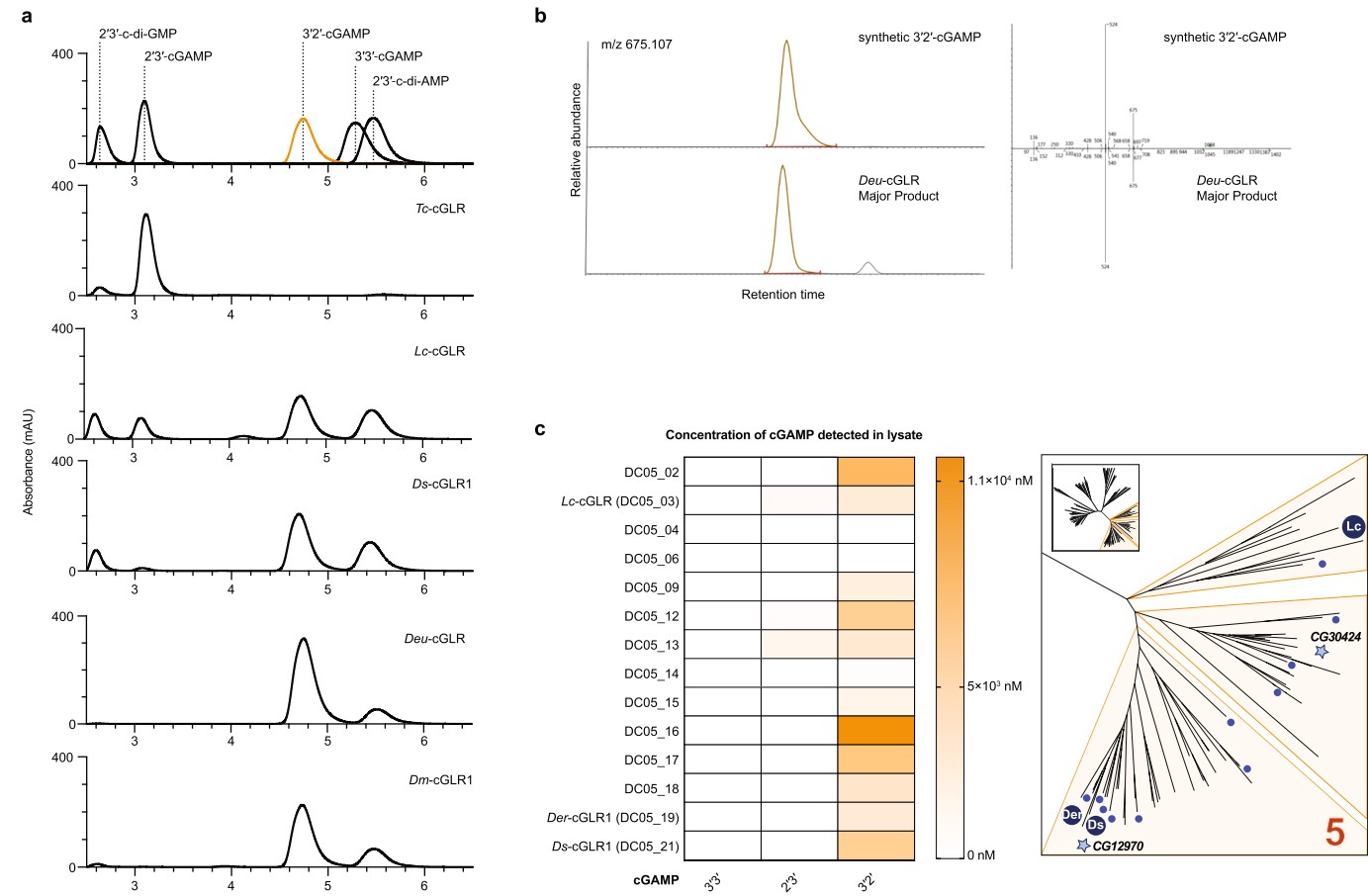

**Extended Data Fig. 6 | Synthesis of 3'2'-cGAMP by Diptera cGLRs. a**, HPLC analysis of the nucleotide products of *Tc*-cGLR, *Dm*-cGLR1, *Ds*-cGLR1, *Lc*-cGLR and *Deu*-cGLR reactions compared with relevant synthetic controls. Integration of major and minor product peaks in *n* = 3 independent experiments was used to calculate relative product ratios shown in Fig. 3c. **b**, The *Drosophila* cGLR major reaction product was purified from *Deu*-cGLR reactions and compared with synthetic 3'2'-cGAMP with tandem mass spectrometry analysis. Parent mass extracted ion trace (left) and tandem mass spectra comparison (right) validate the chemical identity of the *Drosophila*

cGLR product as 3'2'-cGAMP. **c**, Identification of widespread 3'2'-cGAMP synthesis by Diptera cGLRs. The heat map shows the relative concentrations of cGAMP isomers detected by HPLC-MS in bacterial lysates expressing Diptera cGLRs (as described in Extended Data Fig. 2b) (left). In all cases, 3'2'-cGAMP was present as the dominant product with trace amounts of 3'3'-cGAMP and 2'3'-cGAMP detected in some samples as minor species. Right, inset of clade 5 in the Diptera cGLR phylogeny from Fig. 2a annotated to show all enzymes identified to synthesize 3'2'-cGAMP.

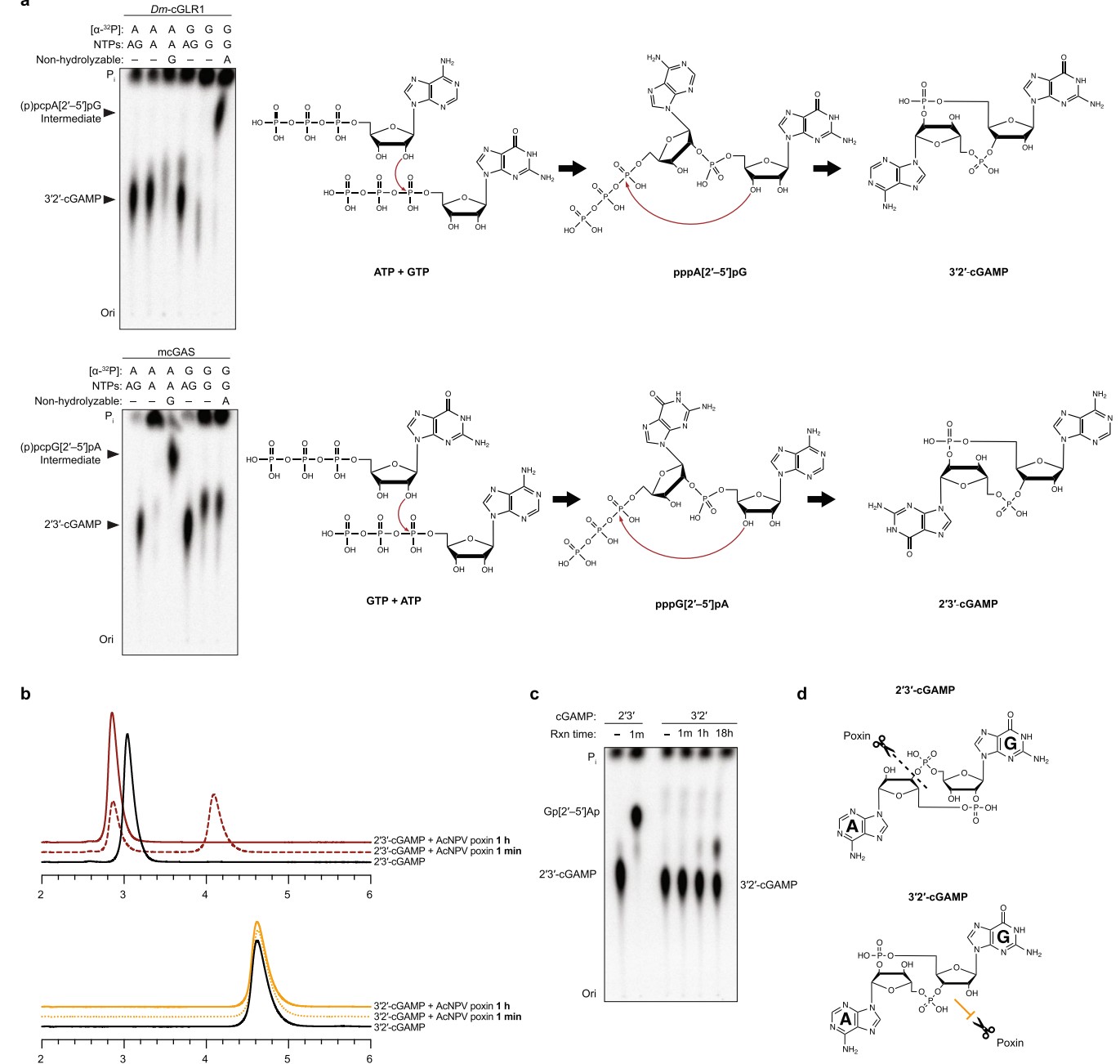

**Extended Data Fig. 7 | Mechanism of 3′2′-cGAMP bond formation and resistance to degradation by viral poxin enzymes. a**, Analysis of *Dm*-cGLR1 reactions with pairwise combinations of α-[32]P-labelled nucleotides and non-hydrolyzable nucleotides reveals reaction intermediates and identifies the order of bond formation during 3′2′-cGAMP synthesis. Left: TLC analysis demonstrates that *Dm*-cGLR1 forms a linear intermediate in the presence of GTP and non-hydrolyzable ATP (Apcpp), indicating that the 2′–5′ phosphodiester bond is synthesized first. Exposed γ-phosphates removed by phosphatase treatment before analysis are indicated in parentheses. Note that while a linear intermediate cannot be formed in the presence of non-hydrolyzable GTP (Gpcpp), *Dm*-cGLR1 will synthesize the off-product 2′3′-c-di-AMP. Mouse cGAS, which synthesizes 2′3′-cGAMP through the linear intermediate pppG[2′–5′]pA, is shown here for comparison[19]. Right: schematic of the reaction mechanism for each enzyme. **b**, Poxins are 2′3′-cGAMP-specific viral nucleases that disrupt cGAS–STING signalling. HPLC analysis of synthetic 2′3′-cGAMP or 3′2′-cGAMP treated with poxin from the insect baculovirus *Autographa californica* nucleopolyhedrovirus (AcNPV) is shown[22,37]. In 1 min, AcNPV poxin cleaves 2′3′-cGAMP into a mixture of intermediate and full-cleavage product; and after 1 h, turnover is complete. No cleavage of 3′2′-cGAMP is observed by AcNPV poxin under these reaction conditions. **c**, Using TLC as a more sensitive assay, we observed minimal cleavage of 3′2′-cGAMP following overnight incubation with AcNPV poxin. Data in **a**–**c** are representative of *n* = 3 independent experiments. **d**, Schematic highlighting how an isomeric switch in phosphodiester linkage specificity makes 3′2′-cGAMP remarkably resistant to poxin-mediated cleavage.

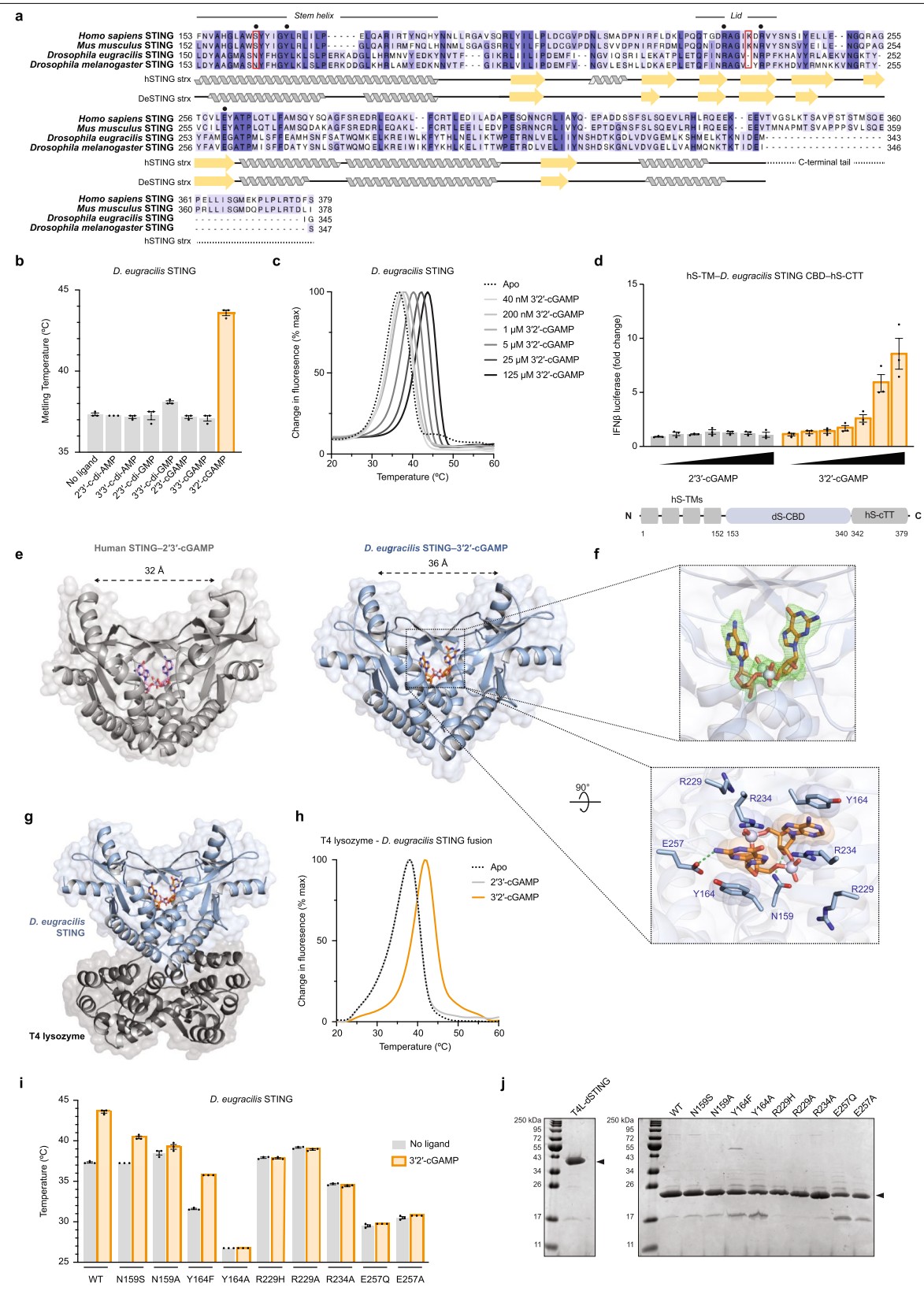

**Extended Data Fig. 8 | See next page for caption.**

**Extended Data Fig. 8 | Structural and biochemical analysis of dSTING.**
**a**, Alignment of the C-terminal CDN-binding domains of human STING, mouse STING, *D. eugracilis* STING and *D. melanogaster* STING. Architecture of the core CDN-binding domain is conserved across metazoans; the disordered C-terminal tail, which controls IRF3–IFNβ signalling, is specific to vertebrates[8,21]. Ligand-interacting residues selected for mutational analysis are denoted with a black circle; Diptera-specific adaptations are highlighted with a red outline. All structural and biochemical experiments were performed with a *D. eugracilis* STING construct terminating at I340. **b**, In vitro thermal denaturation assay analysing dSTING interactions with a panel of CDNs. Only 3′2′-cGAMP forms a thermostable complex with dSTING in vitro (see also Fig. 3d). 2′3′-cGAMP is known to be capable of stimulating dSTING-dependent signalling in vivo[26], supporting that dSTING can engage with 2′3′-cGAMP with lower affinity. This observation is consistent with the weaker recognition of bacteria-derived 3′3′-cGAMP and 3′3′-c-di-GMP by human STING[2,4]. **c**, In vitro thermal denaturation assay demonstrating concentration-dependent thermal shift induced by 3′2′-cGAMP. **d**, Dose titration of 2′3′-cGAMP and 3′2′-cGAMP in human cells demonstrating selective response by dSTING to 3′2′-cGAMP. The *D. eugracilis* CDN-binding domain (CBD) was adapted for downstream signalling in human cells by addition of N-terminal human transmembrane (hTM) domains and the human C-terminal tail (hCTT). **e**, Comparison of the human STING–2′3′-cGAMP and dSTING–3′2′-cGAMP crystal structures reveals a conserved closed homodimer architecture in which apical 'wings' are spread 32–36 Å, demonstrating high-affinity engagement with an endogenous ligand. **f**, Enlarged cutaways of 3′2′-cGAMP in the dSTING crystal structure. Above: the simulated annealing $F_O-F_C$ omit map (contoured at 3 σ). Below: a top-down view highlighting key dSTING–3′2′-cGAMP contacts. **g**, Full crystal structure used to determine the structure of *D. eugracilis* STING in complex with 3′2′-cGAMP. T4 lysozyme is fused to the N terminus of the *D. eugracilis* STING CBD. **h**, Thermal denaturation assay as in Fig. 3d demonstrating that N-terminal fusion of T4 lysozyme does not impair dSTING recognition of 3′2′-cGAMP. **i**, Mutational analysis of key ligand-interacting residues in dSTING; the thermal denaturation assay was used to analyse 3′2′-cGAMP recognition. Mutations that conserve functional contacts with 3′2′-cGAMP (Y164F) maintain ligand recognition; mutations that ablate contacts abrogate ligand binding. N159S exhibits diminished ability to recognize 3′2′-cGAMP. Data in **b** and **i** are mean ± s.e.m. of the average $T_m$ calculated from $n = 2$ technical replicates in $n = 3$ independent experiments. Data in **c** are representative of $n = 3$ independent experiments. Data in **d** are mean ± s.e.m. of $n = 3$ technical replicates and representative of $n = 3$ independent experiments. **j**, SDS–PAGE and Coomassie stain analysis of purified WT and mutant proteins.

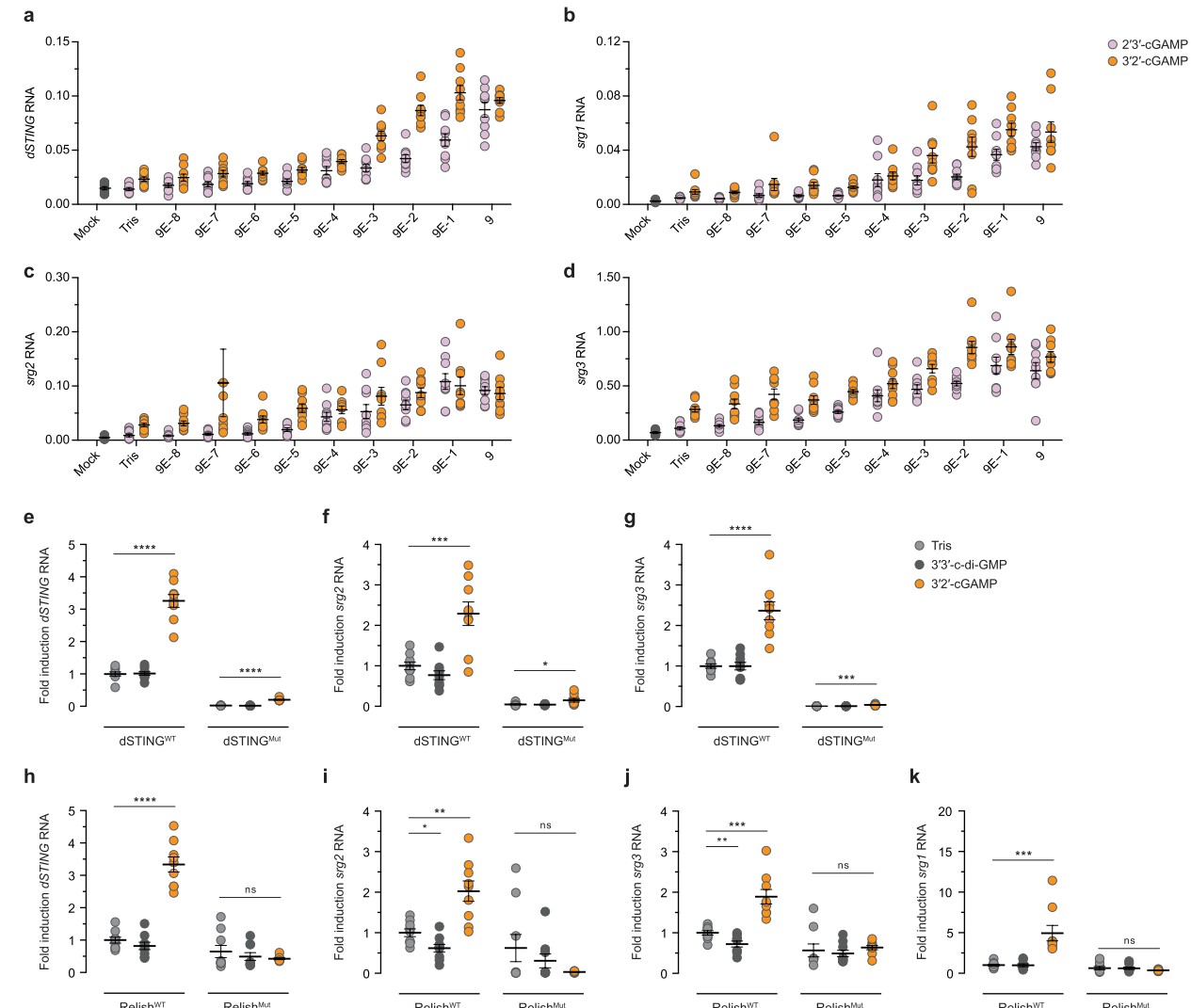

**Extended Data Fig. 9 | 3′2′-cGAMP induces the expression of dSTING-regulated genes. a–d**, Injection of 3′2′-cGAMP into *D. melanogaster* has a dose-dependent effect on the expression of *Sting-regulated genes* (*srgs*). 2′3′-cGAMP was used as positive control as previously characterized[23,26]. Synthetic nucleotide was injected into the body cavity of WT (*w*[1118]) flies and gene expression was measured after 24 h. RNA levels were measured relative to the control gene *RpL32*, and nucleotide concentrations are displayed in µg µl[−1]. Note that for *srg2* measurement after injection of 9E−7 µg µl[−1] 3′2′-cGAMP, there was one outlier replicate with a value of 0.5977 (data not shown, included in

mean analysis). **e–k**, As in Fig. 4a, RNA expression analysis of *Sting-regulated genes* (*srgs*) 24 h after injection with synthetic 3′2′-cGAMP or 3′3′-c-di-GMP. RNA levels are shown as fold induction compared with buffer control in WT flies. dSTING[Mut] = RXN mutant; Relish[Mut] = Relish[E20] mutant, as previously characterized[23,26]. All data in **a–k** represent the mean ± s.e.m. of *n* = 3 independent experiments and each point represents a pool of 6 flies. *P* value ns (>0.05) unless otherwise noted: [****]*P* < 0.0001 (**e**); [***]*P* = 0.0006, [*]*P* = 0.0404 (**f**); [****]*P* < 0.0001, [***]*P* = 0.0002 (**g**); [****]*P* < 0.0001 (**h**); [**]*P* = 0.0015, [*]*P* = 0.0117 (**i**); [***]*P* = 0.0002, [**]*P* = 0.0076 (**j**); [***]*P* = 0.0009 (**k**).

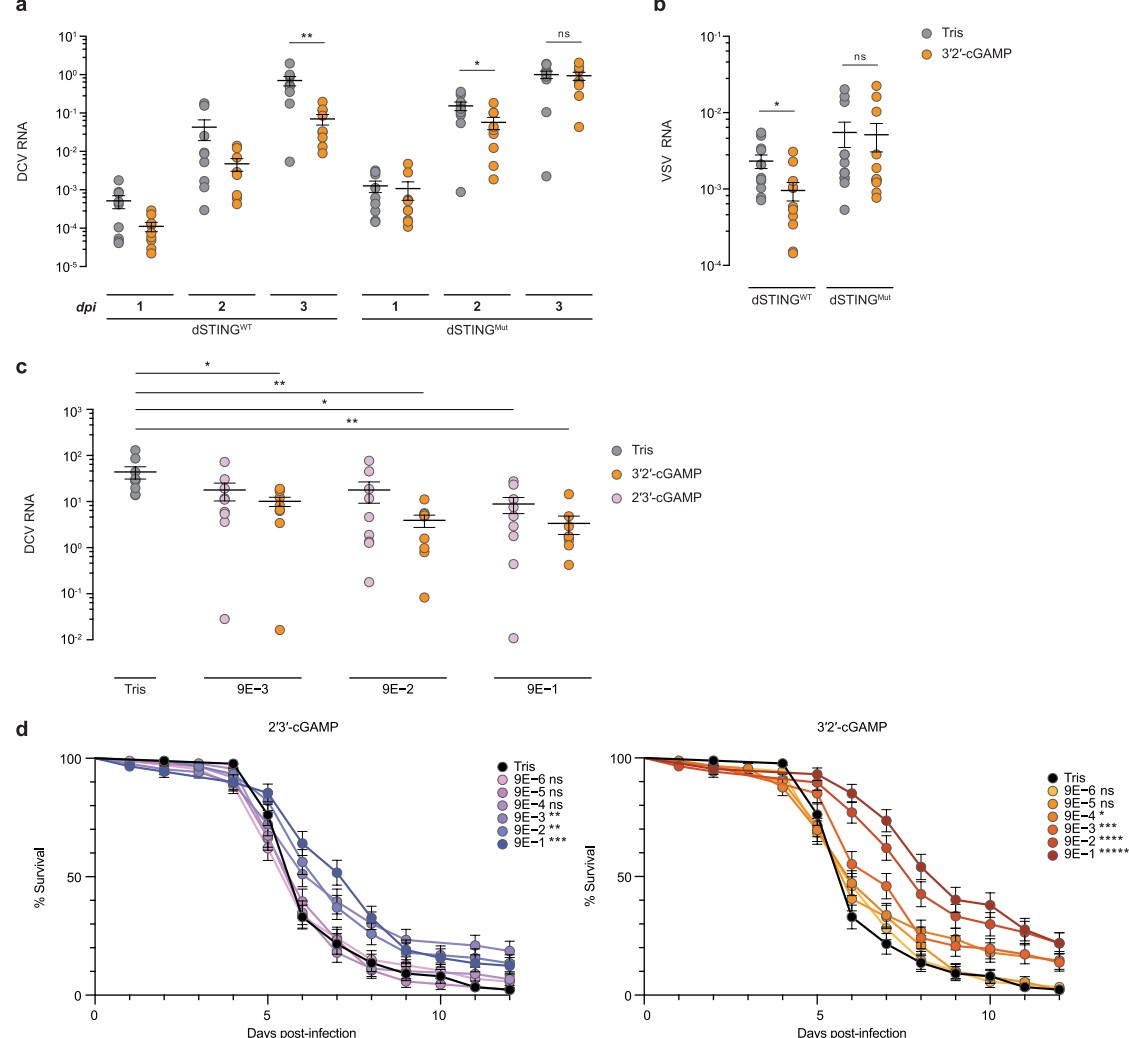

**Extended Data Fig. 10 | 3′2′-cGAMP functions as a potent antiviral ligand.**
**a**, Analysis of the effect of 3′2′-cGAMP on *Drosophila* C virus (DCV) viral RNA load in flies. dSTING WT and mutant flies were co-injected with DCV and 3′2′-cGAMP or buffer control. Viral RNA levels were measured at each time as indicated relative to the control gene *RpL32*. DCV is a picornavirus-like (+) ssRNA virus in the family *Dicistroviridae*. [**]*P* = 0.0051, [*]*P* = 0.0388. **b**, Analysis of the effect of 3′2′-cGAMP on vesicular stomatitis virus (VSV) viral RNA load in flies. dSTING WT and mutant flies were co-injected with VSV and 3′2′-cGAMP or buffer control as in **a**. Viral RNA levels were measured 4 days post-infection (dpi) relative to the control gene *RpL32*. VSV is a (-)ssRNA virus in the *Rhabdoviridae* family. [*]*P* = 0.0185. **c**, Analysis of DCV viral RNA load in flies injected with increasing doses of 3′2′-cGAMP, 2′3′-cGAMP or buffer control (as

in **a**). Viral RNA levels were measured 2 dpi relative to the control gene *RpL32*. For 2′3′-cGAMP injection: 9E−1 [*]*P* = 0.0192. For 3′2′-cGAMP injection: 9E−3 [*]*P* = 0.0212, 9E−2 [**]*P* = 0.0075, 9E−1 [**]*P* = 0.0070. **d**, Survival curves after DCV infection showing the effect of injection with dose titration of 3′2′-cGAMP or 2′3′-cGAMP compared with buffer control. Both cGAMP isomers significantly delay mortality in a dose-dependent manner; 3′2′-cGAMP provides greater protection in comparison to 2′3′-cGAMP. For 2′3′-cGAMP injection: 9E−3 [**]*P* = 0.0047, 9E−2 [**]*P* = 0.0031, 9E−1 [***]*P* = 0.0002. For 3′2′-cGAMP injection: 9E−4 [*]*P* = 0.0344, 9E−3 [***]*P* = 0.0005, 9E−2 [****]*P* < 0.0001, 9E−1 [****]*P* < 0.0001. All data in **a**–**d** represent the mean ± s.e.m. of *n* = 3 independent experiments and each point represents a pool of 6 flies (**a**, **b**) or 10 flies (**c**, **d**). *P* value is ns unless otherwise noted; ns *P* > 0.05.

# Reporting Summary

## Statistics

For all statistical analyses, confirm that the following items are present in the figure legend, table legend, main text, or Methods section.

| n/a | Confirmed | |
|---|---|---|
| ☐ | ☒ | The exact sample size (*n*) for each experimental group/condition, given as a discrete number and unit of measurement |
| ☒ | ☐ | A statement on whether measurements were taken from distinct samples or whether the same sample was measured repeatedly |
| ☐ | ☒ | The statistical test(s) used AND whether they are one- or two-sided<br>*Only common tests should be described solely by name; describe more complex techniques in the Methods section.* |
| ☒ | ☐ | A description of all covariates tested |
| ☒ | ☐ | A description of any assumptions or corrections, such as tests of normality and adjustment for multiple comparisons |
| ☐ | ☒ | A full description of the statistical parameters including central tendency (e.g. means) or other basic estimates (e.g. regression coefficient) AND variation (e.g. standard deviation) or associated estimates of uncertainty (e.g. confidence intervals) |
| ☐ | ☒ | For null hypothesis testing, the test statistic (e.g. *F*, *t*, *r*) with confidence intervals, effect sizes, degrees of freedom and *P* value noted<br>*Give P values as exact values whenever suitable.* |
| ☒ | ☐ | For Bayesian analysis, information on the choice of priors and Markov chain Monte Carlo settings |
| ☒ | ☐ | For hierarchical and complex designs, identification of the appropriate level for tests and full reporting of outcomes |
| ☒ | ☐ | Estimates of effect sizes (e.g. Cohen's *d*, Pearson's *r*), indicating how they were calculated |

*Our web collection on statistics for biologists contains articles on many of the points above.*

## Software and code

Policy information about availability of computer code

| Data collection | All radioactivity-based imaging was collected using Typhoon scanner control 2.0.0.6<br>Chromatography traces collected using GE Unicorn 7.1<br>Protein, DNA, and RNA gel images collected using BioRad ImageLab 2.4.0.3<br>Protein homologs identified using NCBI PSI-BLAST (web-based: https://blast.ncbi.nlm.nih.gov/Blast.cgi) |
|---|---|
| Data analysis | Phenix 1.19, Coot 0.8.9, PyMOL 2.3, GraphPad Prism 9.0.1, Geneious Prime v2020.12.23, Image Quant 8.2.0 |

For manuscripts utilizing custom algorithms or software that are central to the research but not yet described in published literature, software must be made available to editors and reviewers. We strongly encourage code deposition in a community repository (e.g. GitHub). See the Nature Portfolio guidelines for submitting code & software for further information.

## Data

Policy information about availability of data

All manuscripts must include a data availability statement. This statement should provide the following information, where applicable:
- Accession codes, unique identifiers, or web links for publicly available datasets
- A description of any restrictions on data availability
- For clinical datasets or third party data, please ensure that the statement adheres to our policy

Atomic coordinates and structure factors of human MB21D2, T. castaneum cGLR, Drosophila STING, and the Drosophila STING–3'2'-cGAMP complex have been deposited in PDB under the accession codes 7LT1, 7LT2, 7MWY, and 7MWZ. All other data are available in the manuscript or the supplementary materials.

# Field-specific reporting

Please select the one below that is the best fit for your research. If you are not sure, read the appropriate sections before making your selection.

☒ Life sciences ☐ Behavioural & social sciences ☐ Ecological, evolutionary & environmental sciences

For a reference copy of the document with all sections, see nature.com/documents/nr-reporting-summary-flat.pdf

# Life sciences study design

All studies must disclose on these points even when the disclosure is negative.

| Sample size | Sample size for all Drosophila experiments was determined using previously published protocols (Cai et al., 2020, PMID:33262294). |
| --- | --- |
| Data exclusions | No data were excluded from analyses |
| Replication | All experiments were performed with independent replicates as described in the figure legends. |
| Randomization | X-ray crystal structures were refined with a randomly selected R-free reflection set based on automatic selection in Phenix 1.19. Flies were randomly selected for injection with any of the tested CDNs or buffer control. No other randomization was required for the cell biological, biochemical, and structural analyses in this study. |
| Blinding | Blinding was not performed for data analysis or group allocation for Drosophila experiments. Flies were randomly selected for each experimental group and data were collected by unbiased, quantitative means. |

# Reporting for specific materials, systems and methods

We require information from authors about some types of materials, experimental systems and methods used in many studies. Here, indicate whether each material, system or method listed is relevant to your study. If you are not sure if a list item applies to your research, read the appropriate section before selecting a response.

## Materials & experimental systems

| n/a | Involved in the study |
| --- | --- |
| ☒ | ☐ Antibodies |
| ☐ | ☒ Eukaryotic cell lines |
| ☒ | ☐ Palaeontology and archaeology |
| ☐ | ☒ Animals and other organisms |
| ☒ | ☐ Human research participants |
| ☒ | ☐ Clinical data |
| ☒ | ☐ Dual use research of concern |

## Methods

| n/a | Involved in the study |
| --- | --- |
| ☒ | ☐ ChIP-seq |
| ☒ | ☐ Flow cytometry |
| ☒ | ☐ MRI-based neuroimaging |

## Eukaryotic cell lines

Policy information about cell lines

| Cell line source(s) | HEK 293T (catalog ATCC CRL-3216) cells were purchased directly from ATCC. |
| --- | --- |
| Authentication | HEK 293T cells were validated by ATCC |
| Mycoplasma contamination | Cell lines were not tested for mycoplasma contamination. |
| Commonly misidentified lines (See ICLAC register) | No misidentified lines were used. |

## Animals and other organisms

Policy information about studies involving animals; ARRIVE guidelines recommended for reporting animal research

| Laboratory animals | All Drosophila melanogaster fly lines are described in methods, were handled according to standards practices in the field, and are Wolbachia free. Equal numbers of male and female flies were selected for each experimental group. Flies were 3-5 days old at the commencement of each experiment. |
| --- | --- |
| Wild animals | No wild animals were used in this study. |

| | |
|---|---|
| Field-collected samples | No field-collected samples were used in this study. |
| Ethics oversight | No ethics oversight was required for this study. |

Note that full information on the approval of the study protocol must also be provided in the manuscript.

