## [Peer Review File · Nature]

Manuscript Title: cGAS-like receptors sense RNA and control 3'2'-cGAMP signaling in *Drosophila*

Reviewer Comments & Author Rebuttals

Reviewer Reports on the Initial Version:

Referees' comments:

Referee #1 (Remarks to the Author):

Summary of the key results

In this manuscript, Slavik et al. presented a thorough analysis of cGAS-like receptors (cGLRs) from insects and more. Starting with the crystal structure of a beetle cGLR as well as another human cGLR, they show conserved structure around the enzyme active site but altered structure at the ligand binding surface, as well as the activity of the beetle enzyme to generate a c-di-nucleotide in response to dsRNA, as opposed to the DNA responsiveness of the original gGAS. This is followed with a more extensive phylogenetic analysis of cGLRs in insects, which fall into 5 subgroups. Focusing on Group 5, they show many of these cGLRs respond to dsRNA, and demonstrated via point mutation that RNA likely binds in the known ligand binding groove. Importantly, most of these cGLRs generate a novel second messenger, 3'2'-cGAMP, instead of the 2'3'-version generated by DNA-responsive mammalian cGAS; the beetle cGLR is the exception among the insects cGLRs characterized here, generating 2'3'-cGAMP. [It is not clear if this Tc cGLR is in Group 5.] They also show that this alternative 3'2'-cGAMP is resistant to degradation by a viral poxin, hinting at the evolutionary forces driving this expansion. Finally they show that injection of 3'2'-cGAMP into adult flies activates the STING pathway and protects from lethal infection with DCV.

Originality and significance: if not novel, please include reference

The finding of a new second messenger is novel. The ability of cGLR1 to produce this alternative cGAMP is similarly exciting, as is the resistance of the 3'2' molecule to the viral poxin.

Of course, *Drosophila* STING, its responsiveness to (regular) 2'3'-cGAMP, and its antiviral role has been established previously (by some of the authors here), but the novelty of this report lies in the 3'2' responsiveness of insect STING and the unique ability of many insect cGLRs to generate this messenger.

The response of cGLRs to RNA, rather than DNA is highly novel as well, and given the proliferation of this gene family it argues for wider universe of potential ligands for the cGLRs.

Data & methodology: validity of approach, quality of data, quality of presentation

The data is highly robust throughout.

The one area that needs a bit more analysis, is the RNA vs. DNA specificity of cGAS versus Ds-cGLR1. The gel shift in Figure 2f should be performed with the opposite ligands. Or, perhaps, is the nucleic acid binding specificity not so tight, as is hinted at in the condensate assays? If its binding is not so specific to RNA, where does the specificity of the response (to RNA viruses) derive?

Appropriate use of statistics and treatment of uncertainties

Yes, Stats are included throughout and are used correctly.

Conclusions: robustness, validity, reliability

The conclusions are highly robust and most questions have been addressed from multiple angles.

Suggested improvements: experiments, data for possible revision

The one thing missing from this work is the characterization of a cGLR1 mutant (or knockdown) fly. Given the power of genetics and genetic engineering in the *Drosophila melanogaster* model system, this omission really compromises this otherwise robust analysis.

I would suggest moving the poxin analysis into the main body of the data presentation. Biologically, this is highly informative about the likely evolutionary pressures evolved in expanding this family in the insect world.

References: appropriate credit to previous work?

Yes

Clarity and context: lucidity of abstract/summary, appropriateness of abstract, introduction and conclusions

All good.

Referee #2 (Remarks to the Author):

This manuscript supports prior findings on the existence of a functionally relevant cGAS-STING signaling axis in *Drosophila*. Slavik et al utilize structural and bioinformatical tools to demonstrate that insects express several cGAS homologs of which some can be activated by dsRNA. The study of one cGAS-like gene, cGLR1, from *Drosophila melanogaster* revealed that the enzyme responds to long dsRNA and produces an isomer of the cGAS dinucleotide cGAMP contained a distinct mixed-linkage cGAMP molecule. Injection of 3`2`cGAMP into flies induces gene expression in a manner dependent on dSTING and Relish, the *Drosophila* homologue of NF- κ B consistent with prior work (PMID: 30119996) and confers partial protection against two RNA viruses.

Prior reports, including from the Kranzusch and Ilmer lab, have provided evidence on the evolutionary conservation of homologs of cGAS and STING in protecting prokaryotes (or flies) from phage (and viral infection). The presented finding of cGLR1 activation by RNA and 3`2`cGAMP isomer synthesis is novel and the paper is technically sound.

However, more evidence is needed to support the direct importance of cGLR1 and of 3`2`cGAMP in *Drosophila* antiviral immunity. For example, the importance of 3`2`cGAMP is inferred by administration of the dinucleotide into flies. The impact of this experimental manipulation is not very impressive from a biological perspective and a minor advancement compared to prior (and more elaborate) work on cGAMP signaling in *Drosophila* by Goto et al. The authors speculate that the isomer may confer a selective advantage over 2`3`cGAMP owing to resistance to cyclic phosphodiesterase cleavage, however a more in-depth and direct comparison on the distinct effects of the two divergent dinucleotides – both at the level of dSTING binding as well as at the functional consequences – are lacking in the manuscript. The structure of MB21D2 as presented does not add much novelty compared to prior work that characterized human MAB21 like protein, another member of the NTase fold proteins in humans (Oliveira Mann, et al. *Scientific reports* 2016). The mechanistic function, any biochemical activity and interaction partners, ligands or substrates of MB21D2 remain elusive. Hence, the claim of divergent functions of human cGLRs is not further elaborated relative to the state of knowledge.

Given prior publications on the reported connection between cGAMP and STING for *Drosophila* immunity, existence of divergent cyclic dinucleotide isomers, and structural information on metazoan MAB21 proteins, to this reviewer, the manuscript does not yet convey the completeness or novelty that is desired for a publication in *Nature*.

Apart from these general aspects, the authors should clearly show that cGLR1 is selective for dsRNA over dsDNA for protein-nucleic acid complex formation. The claim of cGLR1 phase separating on dsRNA would demand much further validation – the microscopy images per se are

insufficient.

Referee #3 (Remarks to the Author):

In this manuscript, the authors identified from various sources homologues (cGLR) of cGAS, which produces innate immune second messenger 2'3'-cGAMP to activate STING receptor upon pathogen and virus infection. The authors solved the crystal structures of cGLRs from human and insect, which shares structural features with cGAS in RNA binding surface, while the di-nucleotide catalytic pocket is more divergent. Unexpectedly, cGLR synthesizes 3'2'-cGAMP not 2'3'-cGAMP, which also activates STING pathway, but the activating stimuli is unknown. The last point is the topics of this manuscript, and this reviewer claims several major concerns below;

1. The author should clarify the catalytic mechanism of how cGLR synthesizes 3'2'-cGAMP not 2'3'-cGAMP, hopefully by solving the structure of the complex with 3'2'-cGAMP, combined with mutant analysis.

2. About the stimuli for cGLR to synthesize 3'2'-cGAMP, the authors suggest that in addition to cGAS, cGLR has an important role in anti-tumor immunity. The author should provide more evidence for this by additional experiments.

Together, the concept of new second messenger 3'2'-cGAMP synthesized by cGAS homologue, cGLR is quite interesting, but due to lack of experimental data, this topic is still premature or weak. Therefore, this reviewer would like to ask the authors to provide more structural and cell biological data to make this topic more solid for publication in a leading journal of Nature.

Author Rebuttals to Initial Comments:

Referee #1

Summary of the key results

In this manuscript, Slavik et al. presented a thorough analysis of cGAS-like receptors (cGLRs) from insects and more. Starting with the crystal structure of a beetle cGLR as well as another human cGLR, they show conserved structure around the enzyme active site but altered structure at the ligand binding surface, as well as the activity of the beetle enzyme to generate a c-di-nucleotide in response to dsRNA, as opposed to the DNA responsiveness of the original gGAS. This is followed with a more extensive phylogenetic analysis of cGLRs in insects, which fall into 5 subgroups. Focusing on Group 5, they show many of these cGLRs respond to dsRNA, and demonstrated via point mutation that RNA likely binds in the known ligand binding groove. Importantly, most of these cGLRs generate a novel second messenger, 3'2'-cGAMP, instead of the 2'3'-version generated by DNA-responsive mammalian cGAS; the beetle cGLR is the exception among the insects cGLRs characterized here, generating 2'3'-cGAMP. [It is not clear if this Tc cGLR is in Group 5.] They also show that this alternative 3'2'-cGAMP is resistant to degradation by a viral poxin, hinting at the evolutionary forces driving this expansion. Finally they show that injection of 3'2'-cGAMP into adult flies activates the STING pathway and protects from lethal infection with DCV.

Originality and significance: if not novel, please include reference

The finding of a new second messenger is novel. The ability of cGLR1 to produce this alternative cGAMP is similarly exciting, as is the resistance of the 3'2' molecule to the viral poxin.

*Of course, *Drosophila* STING, its responsiveness to (regular) 2'3'-cGAMP, and its antiviral role has been established previously (by some of the authors here), but the novelty of this report lies in the 3'2' responsiveness of insect STING and the unique ability of many insect cGLRs to generate this messenger.*

The response of cGLRs to RNA, rather than DNA is highly novel as well, and given the proliferation of this gene family it argues for wider universe of potential ligands for the cGLRs.

We appreciate the reviewer for highlighting our work as exciting and thorough, and we thank them for their helpful comments to improve our manuscript. We are glad the reviewer highlights the discovery of 3'2'-cGAMP as novel and point them to our new findings that explain the structural basis of *Drosophila* STING 3'2'-cGAMP-recognition and demonstrate 3'2'-cGAMP protects flies from viral infection more potently than 2'3'-cGAMP (see response to “Suggested Improvements” below).

To clarify the relationship between insect cGLR enzymes presented in our study we now include a supplementary table listing the full sequence accession and “Clade” designation for each protein (Supplementary Table 2). *Tc*-cGLR is part of an outgroup of related insect cGLRs and does not appear on the tree in Figure 2a focused on cGLR enzymes in the order *Diptera*.

Data & methodology: validity of approach, quality of data, quality of presentation

The data is highly robust throughout.

*The one area that needs a bit more analysis, is the RNA vs. DNA specificity of cGAS versus *Ds*-cGLR1. The gel shift in Figure 2f should be performed with the opposite ligands. Or, perhaps, is the nucleic acid binding specificity not so tight, as is hinted at in the condensate assays? If it binding is not so specific to RNA, where does the specificity of the response (to RNA viruses) derive?*

We have repeated the *Ds*-cGLR1 and human cGAS EMSAs with opposite ligands and include these new data in Extended Data Fig. 5a. Our results further establish that *Ds*-cGLR1 interacts with dsRNA with higher affinity than dsDNA and agree with our biochemical data demonstrating that *Drosophila* cGLR1 enzymes are activated exclusively in response to long >30 bp double-stranded RNA. Similar to previous observations measuring off-pathway interaction between human cGAS and dsRNA (Civril et al. *Nature* 2013 PMID 23722159), we also observe residual interaction between cGLR1 and dsDNA likely due to non-specific charged interactions. However, cGLR1 interactions with dsDNA are incapable of driving enzyme activation and nucleotide second messenger synthesis (Extended Data Fig. 4a).

As additional data in our revised manuscript, we adapted a HEK293T luciferase reporter system we previously used to characterize mammalian cGAS (Kranzusch et al, *Cell Reports* 2013 PMID 23707061) to reconstitute *Drosophila* cGLR1 activity in cells. We show that *Dm*-cGLR1 and *Ds*-cGLR1 activation of a STING-dependent response is strictly reliant on dsRNA stimulation (Fig. 2h; Extended Data Fig. 4e). Mutations to the *Dm*-cGLR1 and *Ds*-cGLR1 catalytic active site or RNA-binding groove disrupt signaling and prevent downstream STING activation (Fig. 2h; Extended Data Fig. 3f), in agreement with

our *in vitro* biochemical findings (Fig. 2g; Extended Data Fig. 3b–e). Our data explain how cGAS and cGLR enzymes sense ligands through higher-order complex formation and demonstrate that only correct ligand recognition can drive activation of specific nucleotide second messenger synthesis.

Appropriate use of statistics and treatment of uncertainties

Yes, Stats are included throughout and are used correctly.

We thank the reviewer for verifying our statistical methods.

Conclusions: robustness, validity, reliability

The conclusions are highly robust and most questions have been addressed from multiple angles.

We thank the reviewer for their comment that our conclusions are highly robust.

Suggested improvements: experiments, data for possible revision

*The one thing missing from this work is the characterization of a cGLR1 mutant (or knockdown) fly. Given the power of genetics and genetic engineering in the *Drosophila melanogaster* model system, this omission really compromises this otherwise robust analysis.*

We agree with the reviewer that our discoveries of *Drosophila* cGLR1 as a dsRNA-sensor and 3'2'-cGAMP as a novel nucleotide second messenger provide the foundation for *in vivo* studies to further explain the regulation of this pathway in animals. In our manuscript, we use *in vivo* studies to focus on defining the role of the cGLR1 product 3'2'-cGAMP as a second messenger that signals through a STING-Relish(NF- κ B) axis to protect animals from RNA viral challenge (Fig. 5a–d; Extended Data Fig. 9 and 10).

We point the reviewer to a series of new experiments included to improve the robustness of our analysis. In our revised manuscript we present new experiments directly comparing the ability of 3'2'-cGAMP and 2'3'-cGAMP to activate *Drosophila* STING signaling in cells and *in vivo*. A dose titration of 3'2'-cGAMP and 2'3'-cGAMP in cells reveals that *Drosophila* STING selectively responds to 3'2'-cGAMP and that 2'3'-cGAMP does not induce a signaling response in the tested range of concentrations in this assay (Extended Data Fig. 8d). We further use viral challenge experiments in *Drosophila melanogaster* to compare the function of 3'2'-cGAMP and 2'3'-cGAMP as antiviral second messengers *in vivo* (Fig. 5d; Extended Data Fig. 10c,d). While 2'3'-cGAMP provides protection from RNA viral infection, 3'2'-cGAMP more potently suppresses RNA viral replication and increases fly survival in a dose-dependent manner. Together, our results demonstrate that *Drosophila* STING selectively responds to 3'2'-cGAMP and more weakly interacts with 2'3'-cGAMP, similar to weaker recognition of bacteria-derived 3'3'-cGAMP and 3'3'-c-di-GMP by human STING (Burdette et al. *Nature* 2011 PMID 21947006; Yin et al. *Molecular Cell* 2012 PMID 22705373; Gao et al. *Cell* 2013 PMID 23910378).

We additionally provide new structural and biochemical data to directly compare the ability of 3'2'-cGAMP and 2'3'-cGAMP to bind *Drosophila* STING. First, as a major breakthrough, we have determined a 1.8 Å crystal structure of *Drosophila eugracilis* STING and a 2.0 Å crystal structure of the *Drosophila* STING–3'2'-cGAMP complex (Fig. 4b–f; Extended Data Fig. 8e–g; Supplementary Table 1). Insect STING proteins have been refractory to biochemical experiments and this technical barrier has limited functional understanding of STING signaling in *Drosophila* (Kranzusch et al. *Molecular Cell* 2015 PMID 26300263; Martin et al. *Cell Reports* 2018 PMID 29924997; Goto et al. *Immunity* 2018 PMID 30119996). Through screening a large number of *Drosophila* STING homologs and purification conditions we have now overcome this barrier and report a full structural and biochemical characterization of insect STING. Our new structural and supporting biochemical data explain how adaptation of key ligand binding residues and a single amino acid deletion in the *Drosophila* STING β-strand lid enable specific recognition of 3'2'-cGAMP (Fig. 4b–f; Extended Data Fig. 8). We confirm that *Drosophila* STING selectively recognizes 3'2'-cGAMP *in vitro* (Fig. 4a; Extended Data Fig. 8b) and validate the structural findings with a panel of mutant *Drosophila* STING proteins demonstrating that only correct recognition of 3'2'-cGAMP results in formation of a thermo-stable complex (Extended Data Fig. 8i,j).

Together our data link biological insights from the molecular to the organismal level to explain a novel antiviral signaling system in a key model of metazoan immunity. Significantly, our characterization of *Drosophila* cGLR1 reveals the first nucleic acid sensing pattern recognition receptor in *Drosophila melanogaster*, overturning previous dogma that *Drosophila* and other insects detect viral RNA solely through RNA-interference. These data also provide the first evidence that animal cGLRs can sense ligands other than double-stranded DNA; previous efforts have been unable to identify the PAMP of any invertebrate cGAS homolog. While prior work has demonstrated an antiviral effect for 2'3'-cGAMP in *Drosophila* (Cai et al. *Science Signaling* 2020 PMID 33262294), our manuscript reveals that the novel isomer 3'2'-cGAMP is synthesized through an endogenous RNA-sensing pathway and is a more potent antiviral ligand for *Drosophila* STING. This discovery provides the first evidence that CDNs beyond 2'3'-cGAMP function as endogenous second messengers in metazoans.

I would suggest moving the poxin analysis into the main body of the data presentation. Biologically, this is highly informative about the likely evolutionary pressures evolved in expanding this family in the insect world.

We agree that our poxin analysis experiments are particularly important and we thank the reviewer for this suggestion. However, due to figure space constraints we have elected to include the new *Drosophila* STING structural data in the main text and leave the poxin analysis experiment in the Extended Data.

References: appropriate credit to previous work?

Yes

Clarity and context: lucidity of abstract/summary, appropriateness of abstract, introduction and conclusions

All good.

We thank the reviewer for verifying our references and the clarity of our text.

Referee #2

This manuscript supports prior findings on the existence of a functionally relevant cGAS-STING signaling axis in Drosophila. Slavik et al utilize structural and bioinformatical tools to demonstrate that insects express several cGAS homologs of which some can be activated by dsRNA. The study of one cGAS-like gene, cGLR1, from Drosophila melanogaster revealed that the enzyme responds to long dsRNA and produces an isomer of the cGAS dinucleotide cGAMP contained a distinct mixed-linkage cGAMP molecule. Injection of 3`2`cGAMP into flies induces gene expression in a manner dependent on dSTING and Relish, the drosophila homologue of NF-kB consistent with prior work (PMID: 30119996) and confers partial protection against two RNA viruses.

Prior reports, including from the Kranzusch and Ilmer lab, have provided evidence on the evolutionary conservation of homologs of cGAS and STING in protecting prokaryotes (or flies) from phage (and viral infection). The presented finding of cGLR1 activation by RNA and 3`2` cGAMP isomer synthesis is novel and the paper is technically sound.

We are glad the reviewer found our data to be novel and technically sound and we thank them for their helpful comments to further improve our manuscript. While previous work has demonstrated an important role for *Drosophila* STING in antiviral immunity, the activating PAMP, upstream pattern-recognition receptor, and nucleotide messenger controlling this signaling axis in flies were unknown prior to this study.

However, more evidence is needed to support the direct importance of cGLR1 and of 3`2`cGAMP in drosophila antiviral immunity. For example, the importance of 3`2`cGAMP is inferred by administration of the dinucleotide into flies. The impact of this experimental manipulation is not very impressive from a biological perspective and a minor advancement compared to prior (and more elaborate) work on cGAMP signaling in drosophila by Goto et al. The authors speculate that the isomer may confer a selective advantage over 2`3`cGAMP owing to resistance to cyclic phosphodiesterase cleavage, however a more in-depth and direct comparison on the distinct effects of the two divergent dinucleotides – both at the level of dSTING binding as well as at the functional consequences - are lacking in the manuscript.

In our revised manuscript we present a series of new structural, cell biological, and *in vivo* experiments to support our discovery of *Drosophila* cGLR1 as a dsRNA sensor that synthesizes the novel antiviral CDN 3`2`-cGAMP and highlight our new findings that 3`2`-cGAMP is a specific and potent activator of *Drosophila* STING.

In support of our results demonstrating that *Drosophila* cGLR1 responds to dsRNA, we adapted a HEK293T luciferase reporter system we previously used to characterize mammalian cGAS (Kranzusch et al, *Cell Reports* 2013 PMID 23707061) to reconstitute *Drosophila* cGLR1 activity in cells. We show that *Dm*-cGLR1 and *Ds*-cGLR1 activation of a STING-dependent response is strictly reliant on dsRNA stimulation (Fig. 2h; Extended Data Fig. 3f) and that this activity correlates with dsRNA concentration (Extended Data Fig. 4e). Mutations to the *Dm*-cGLR1 and *Ds*-cGLR1 catalytic active site or RNA-binding groove disrupt signaling and prevent downstream STING activation (Fig. 2h; Extended Data Fig. 3f) in agreement with our *in vitro* biochemical findings (Fig. 2g; Extended Data Fig. 3b–e).

As requested by the reviewer we provide new structural and biochemical data to directly compare the ability of 2'3'-cGAMP and 3'2'-cGAMP to bind *Drosophila* STING. First, as a major breakthrough, we have determined a 1.8 Å crystal structure of *Drosophila eugracilis* STING and a 2.0 Å crystal structure of the *Drosophila* STING–3'2'-cGAMP complex (Fig. 4b–f; Extended Data Fig. 8e–g; Supplementary Table 1). Insect STING proteins have been refractory to biochemical experiments and this technical barrier has limited functional understanding of STING signaling in *Drosophila* (Kranzusch et al. *Molecular Cell* 2015 PMID 26300263; Martin et al. *Cell Reports* 2018 PMID 29924997; Goto et al. *Immunity* 2018 PMID 30119996). Through screening a large number of *Drosophila* STING homologs and purification conditions we have now overcome this barrier and report a full structural and biochemical characterization of insect STING. Our new structural and supporting biochemical data explain how adaptation of key ligand binding residues and an amino-acid deletion in the *Drosophila* STING β-strand lid enable specific recognition of 3'2'-cGAMP (Fig. 4b–f; Extended Data Fig. 8). Using a thermo-fluor assay we confirm that *Drosophila* STING selectively recognizes 3'2'-cGAMP *in vitro* (Fig. 4a; Extended Data Fig. 8b). We validate these findings with a panel of mutant *Drosophila* STING proteins and demonstrate that only correct recognition of 3'2'-cGAMP results in formation of a thermo-stable complex (Extended Data Fig. 8i,j).

To further extend our biochemical findings we also directly compare 3'2'-cGAMP and 2'3'-cGAMP signaling in cells and *in vivo* as requested. A dose titration of 3'2'-cGAMP and 2'3'-cGAMP in cells confirms that *Drosophila* STING selectively responds to 3'2'-cGAMP and that 2'3'-cGAMP does not induce a signaling response in the tested range of concentrations (Extended Data Fig. 8d). Finally, we use viral challenge experiments in *Drosophila melanogaster* to compare the function of 3'2'-cGAMP and 2'3'-cGAMP as antiviral second messengers *in vivo* (Fig. 5d; Extended Data Fig. 10c,d). While 2'3'-cGAMP provides some protection from RNA viral infection, consistent with previous findings by the Imler lab (Cai et al. *Science Signaling* 2020 PMID 33262294), 3'2'-cGAMP more potently suppresses RNA viral replication and increases fly survival in a dose-dependent manner. Together, our results demonstrate that *Drosophila* STING selectively responds to 3'2'-cGAMP and more weakly interacts with 2'3'-cGAMP, similar to weaker recognition of bacteria-derived 3'3'-cGAMP and 3'3'-c-di-GMP by human STING (Burdette et al. *Nature* 2011 PMID 21947006; Yin et al. *Molecular Cell* 2012 PMID 22705373; Gao et al. *Cell* 2013 PMID 23910378).

The structure of MB21D2 as presented does not add much novelty compared to prior work that characterized human MAB21 like protein, another member of the NTase fold proteins in humans (Oliveira Mann, et al. Scientific reports 2016). The mechanistic function, any biochemical activity and interaction partners, ligands or substrates of MB21D2 remain elusive. Hence, the claim of divergent functions of human cGLRs is not further elaborated relative to the state of knowledge.

In addition to the importance of our work to general understanding of animal innate immunity, our structural results with MB21D2 specifically extend our findings to human cell biology. Our structure of human MB21D2 reveals the first cGAS homolog in humans that is structurally competent for nucleotide second messenger synthesis (Fig. 1a; Extended Data Fig. 1b, Supplementary Table 1). As noted by de Oliveira Mann et al., the structure of human Mab21L1 demonstrates that this protein lacks the catalytic triad necessary for catalysis and cannot function as an innate immune sensor (de Oliveira Mann et al. *Scientific Reports* 2016 PMID 27271801). Furthermore, our manuscript provides the first description of how structural remodeling of the cGLR ligand binding groove enables divergent ligand recognition. The human MB21D2 structure reveals significant alterations to this region, demonstrating that unlike human cGAS and *Drosophila* cGLR1, MB21D2 recognizes a ligand distinct from nucleic acid (Fig. 1a,b). As new data in our revised manuscript we further confirm these findings and demonstrate that MB21D2 is not activated in the presence of nucleic acid or a diverse panel of 10 agonists known to stimulate human innate immunity (Extended Data Fig. 1d,e). Although the stimulating ligand of MB21D2 remains unknown, our structural results demonstrating MB21D2 is capable of nucleotide second messenger synthesis provide a new explanation for why this enzyme is frequently mutated in cancer and create a foundation for characterizing the role of MB21D2 and other cGLRs in human biology.

Given prior publications on the reported connection between cGAMP and STING for drosophila immunity, existence of divergent cyclic dinucleotide isomers, and structural information on metazoan MAB21 proteins, to this reviewer, the manuscript does not yet convey the completeness or novelty that is desired for a publication in Nature.

In addition to our new experimental data, we have updated the text to improve clarity of why our findings represent a key discovery in animal innate immunity (see Lines 158–181). Together our data link biological insights from the molecular to the organismal level to explain a novel antiviral signaling system in a key model of metazoan immunity. Significantly, our characterization of *Drosophila* cGLR1 reveals the first nucleic acid sensing pattern recognition receptor in *Drosophila melanogaster*, overturning previous dogma that *Drosophila* and other insects detect viral RNA solely through RNA-interference. These data also provide the first evidence that animal cGLRs can sense ligands other than double-stranded DNA; previous efforts have been unable to identify the PAMP of any invertebrate cGAS homolog. While prior work has demonstrated an antiviral effect for 2'3'-cGAMP in *Drosophila* (Cai et al. *Science Signaling* 2020 PMID 33262294), our manuscript reveals that the novel isomer 3'2'-cGAMP is synthesized through an endogenous pathway and is a more potent antiviral ligand for *Drosophila* STING. This discovery provides the first evidence that CDNs beyond 2'3'-cGAMP function as endogenous second messengers in metazoans.

Apart from these general aspects, the authors should clearly show that cGLR1 is selective for dsRNA over dsDNA for protein-nucleic acid complex formation. The claim of cGLR1 phase separating on dsRNA would demand much further validation – the microscopy images per se are insufficient.

We have repeated the *Ds*-cGLR1 and human cGAS EMSAs with alternative ligands and now include these data in Extended Data Fig. 5a. Our results further establish that *Ds*-cGLR1 interacts with dsRNA with higher affinity than dsDNA and agree with our biochemical data demonstrating that *Drosophila* cGLR1 enzymes are activated exclusively in response to long >30 bp double-stranded RNA. Similar to previous observations measuring off-pathway interaction between human cGAS and dsRNA (Civril et al. *Nature* 2013 PMID 23722159), we also observe residual interaction between cGLR1 and dsDNA

likely due to non-specific charged interactions. However, cGLR1 interactions with dsDNA are incapable of driving enzyme activation and nucleotide second messenger synthesis (Extended Data Fig. 3f). All of our data are consistent with a model of where cGAS and cGLR enzymes sense ligands through higher-order complex formation and that only correct ligand recognition can drive activation of specific nucleotide second messenger synthesis. We agree with the reviewer regarding dsRNA-driven phase separation of cGLR1 and have reworded Lines 64–67 to describe higher-order complexes as “condensates.” We have moved the microscopy images demonstrating that *Ds*-cGLR1 condensate formation is selectively driven by dsRNA to Extended Data Fig. 5b,c.

Referee #3

In this manuscript, the authors identified from various sources homologues (cGLR) of cGAS, which produces innate immune second messenger 2'3'-cGAMP to activate STING receptor upon pathogen and virus infection. The authors solved the crystal structures of cGLRs from human and insect, which shares structural features with cGAS in RNA binding surface, while the di-nucleotide catalytic pocket is more divergent. Unexpectedly, cGLR synthesizes 3'2'-cGAMP not 2'3'-cGAMP, which also activates STING pathway, but the activating stimuli is unknown. The last point is the topics of this manuscript, and this reviewer claims several major concerns below;

We are glad the reviewer found our results quite interesting, and we thank them for their helpful feedback to further improve our manuscript.

1. The author should clarify the catalytic mechanism of how cGLR synthesizes 3'2'-cGAMP not 2'3'-cGAMP, hopefully by solving the structure of the complex with 3'2'-cGAMP, combined with mutant analysis.

Our results demonstrate that *Drosophila* cGLR1 synthesizes 3'2'-cGAMP because of a change in the order that the ATP and GTP substrate nucleobases are coordinated in the enzyme donor and acceptor pockets (Extended Data Fig. 7a). Human cGAS first uses ATP as a donor nucleotide and produces the reaction intermediate pppG[2'–5']pA prior to cyclization and release of 2'3'-cGAMP (Gao et al. *Cell* 2013 PMID 23647843). In contrast, our results mechanistically explain that *Drosophila* cGLR1 first uses GTP as a donor nucleotide and produces the reaction intermediate pppA[2'–5']pG prior to cyclization and release of 3'2'-cGAMP (Extended Data Fig. 7a).

We have attempted to crystallize *Drosophila* cGLR1 in complex with dsRNA and nucleotide substrates as requested. In spite of extensive trials, we have not been able to identify conditions for crystallization of the cGLR1–dsRNA complex. Our biochemistry data demonstrate that *Drosophila* cGLR1 forms a higher-order assembly in the presence of dsRNA similar to the higher-order human cGAS–dsDNA complex (Fig. 2e; Extended Data Fig. 5a,b,c) and we note that determination of the structure of human cGAS in an active conformation bound DNA was a >6-year effort in the field that required extensive genetic mapping and protein engineering (Zhou et al. *Cell* 2018 PMID 30007416; Xie et al. *PNAS* 2019 PMID 31142647). We therefore have additionally focused on acquiring new *Drosophila* STING

structural biology data to further explain the molecular basis of 3'2'-cGAMP signaling and extend the findings of our manuscript (see response to the final reviewer point below).

2. About the stimuli for cGLR to synthesize 3'2'-cGAMP, the authors suggest that in addition to cGAS, cGLR has important role in anti-tumor immunity. The author should provide more evidence for this by additional experiments.

Our manuscript provides extensive data demonstrating that double-stranded RNA activates *Drosophila* cGLR1 to synthesize 3'2'-cGAMP. Using a minimal *in vitro* system, we demonstrate that dsRNA directly induces the enzymatic activity of numerous insect cGLRs, including from the beetle *Tribolium castaneum* (Fig. 1c; Extended Data Fig. 4a) and from the dipteran flies *Drosophila eugracilis*, *Lucilia cuprina*, *Drosophila erecta*, *Drosophila simulans*, and *Drosophila melanogaster* (Fig. 2b,c; Extended Data Fig. 4a). As new data in our revised manuscript, we leverage these biochemical findings to confirm that *Drosophila* cGLR1 functions as a dsRNA-sensor in cells (Fig. 2h; Extended Data Fig. 3f and 4e). Through a detailed biochemical analysis, we discover that dsRNA-stimulation of *Drosophila* cGLR1 controls the specific synthesis of the novel CDN product 3'2'-cGAMP and that 3'2'-cGAMP is widely conserved as a second messenger in the order *Diptera* (Fig. 3a–d; Extended Data Fig. 6a–c). In contrast, analysis of the *Tribolium castaneum* cGLR product demonstrates synthesis of 2'3'-cGAMP suggesting divergent cGLRs synthesize distinct cyclic dinucleotide signals and that 3'2'-cGAMP is specific to *Dipteran* insects (Fig. 3d; Extended Data Fig. 6a). New data in this revised manuscript further demonstrate that 3'2'-cGAMP is a specific and potent agonist of *Drosophila* STING (Fig. 4a–f; Extended Data Fig. 8b–i). Thus our *in vitro* discovery of both the activating PAMP and the nucleotide product of *Drosophila* cGLR1 provide the foundation to understand the cGLR1-STING-NFκB signaling axis in animals. Working with Jean-Luc Imler and Hua Cai, we show *in vivo* in *Drosophila* that 3'2'-cGAMP signals through STING and Relish to protect flies from RNA viral infection and that 3'2'-cGAMP is a more potent antiviral signal than 2'3'-cGAMP (Fig. 5a–d; Extended Data Fig. 9 and 10). These data provide the context to understand previous findings on the importance of STING in mediating *Drosophila* immunity to RNA viral infection (Martin et al. *Cell Reports* 2018 PMID 29924997; Goto et al. *Immunity* 2018 PMID 30119996; Cai et al. *Science Signaling* 2020 PMID 33262294).

Although the major focus on our manuscript is the discovery of dsRNA sensing in *Drosophila* controlled by the cGLR1–3'2'-cGAMP–STING axis, we use additional structural characterization of the human protein MB21D2 to reveal the breadth of diverse cGLR-family enzymes in metazoans (Fig. 1a,b). While the stimulating ligand of MB21D2 remains unknown (Extended Data Fig. 1d,e), our structural results demonstrate MB21D2 is capable of nucleotide second messenger synthesis and provide a new potential explanation for why this enzyme is frequently mutated in cancer. We have corrected the sentence in our discussion to remove the statement about antitumor immunity and instead focus on highlighting how our structural analysis creates a foundation for characterizing the role of MB21D2 and other cGLRs in human biology (see Lines 171–177).

Together, the concept of new second messenger 3'2'-cGAMP synthesized by cGAS homologue, cGLR is quite interesting, but due to lack of experimental data, this topics is still premature or weak. Therefore, this reviewer would like to ask the authors to provide more structural and cell biological data to make this topic more solid for publication in leading journal of Nature.

We thank the reviewer for their interest in our discovery of 3'2'-cGAMP as a novel antiviral second messenger produced by *Drosophila* cGLR1 in response to dsRNA. To address the reviewer's request for further data to support these findings, we present in the revised manuscript new structural, biochemical, cell biological, and *in vivo* data to complement our prior analysis.

In support of our results demonstrating that *Drosophila* cGLR1 responds to dsRNA, we adapted a HEK293T luciferase reporter system we previously used to characterize mammalian cGAS (Kranzusch et al, *Cell Reports* 2013 PMID 23707061) to reconstitute *Drosophila* cGLR1 activity in cells. These experiments demonstrate that *Dm*-cGLR1 and *Ds*-cGLR1 activation of a STING-dependent response is strictly reliant on dsRNA stimulation (Fig. 2h; Extended Data Fig. 3f and 4e). Mutations to the *Dm*-cGLR1 and *Ds*-cGLR1 catalytic active site or RNA-binding groove disrupt signaling and prevent downstream STING activation (Fig. 2h; Extended Data Fig. 3f), in agreement with our *in vitro* biochemical findings (Fig. 2g; Extended Data Fig. 3b–e).

We also provide new structural and biochemical data to directly compare the ability of 2'3'-cGAMP and 3'2'-cGAMP to bind *Drosophila* STING. First, as a major breakthrough, we have determined a 1.8 Å crystal structure of *Drosophila eugracilis* STING and a 2.0 Å crystal structure of the *Drosophila* STING–3'2'-cGAMP complex (Fig. 4b–f; Extended Data Fig. 8e–g; Supplementary Table 1). Insect STING proteins have been refractory to biochemical experiments and this technical barrier has limited functional understanding of STING signaling in *Drosophila* (Kranzusch et al. *Molecular Cell* 2015 PMID 26300263; Martin et al. *Cell Reports* 2018 PMID 29924997; Goto et al. *Immunity* 2018 PMID 30119996). Through screening a large number of *Drosophila* STING homologs and purification conditions we have now overcome this barrier and report a full structural and biochemical characterization of insect STING. Our new structural and supporting biochemical data explain how adaptation of key ligand binding residues and a single amino acid deletion in the *Drosophila* STING β-strand lid enable specific recognition of 3'2'-cGAMP (Fig. 4b–f; Extended Data Fig. 8). Using a thermo-fluor assay we confirm that *Drosophila* STING selectively recognizes 3'2'-cGAMP and displays no detectable recognition of other CDNs *in vitro* (Fig. 4a; Extended Data Fig. 8b). We validate the structural findings with a panel of mutant *Drosophila* STING proteins and demonstrate that only correct recognition of 3'2'-cGAMP results in formation of a thermo-stable complex (Extended Data Fig. 8i).

To further extend our biochemical findings we also directly compare 3'2'-cGAMP and 2'3'-cGAMP signaling in cells and *in vivo*. A dose titration of 3'2'-cGAMP and 2'3'-cGAMP in cells confirms that *Drosophila* STING selectively responds to 3'2'-cGAMP and that 2'3'-cGAMP does not induce a signaling response in the tested range of concentrations (Extended Data Fig. 8d). Finally, we use viral challenge experiments in *Drosophila melanogaster* to compare the function of 3'2'-cGAMP and 2'3'-cGAMP as antiviral second messengers *in vivo* (Fig. 5d; Extended Data Fig. 10c,d). While 2'3'-cGAMP provides some protection from RNA viral infection, consistent with previous findings by the Imler lab (Cai et al. *Science Signaling* 2020 PMID 33262294), 3'2'-cGAMP more potently suppresses RNA viral replication and increases fly survival in a dose-dependent manner. Together, our results demonstrate that *Drosophila* STING selectively responds to 3'2'-cGAMP and more weakly interacts with 2'3'-cGAMP, similar to the weaker recognition of bacteria-derived 3'3'-cGAMP and 3'3'-c-di-GMP by human STING (Burdette et al. *Nature* 2011 PMID 21947006; Yin et al. *Molecular Cell* 2012 PMID 22705373; Gao et al. *Cell* 2013 PMID 23910378).

Together our data link biological insights from the molecular to the organismal level to explain a novel antiviral signaling system in a key model of metazoan immunity. Significantly, our characterization of *Drosophila* cGLR1 reveals the first nucleic acid sensing pattern recognition receptor in *Drosophila melanogaster*, overturning previous dogma that *Drosophila* and other insects detect viral RNA solely through RNA-interference. These data also provide the first evidence that animal cGLRs can sense ligands other than double-stranded DNA; previous efforts have been unable to identify the PAMP of any invertebrate cGAS homolog. While prior work has demonstrated an antiviral effect for 2'3'-cGAMP in *Drosophila* (Cai et al. *Science Signaling* 2020 PMID 33262294), our manuscript reveals that the novel isomer 3'2'-cGAMP is synthesized through an endogenous pathway and is a more potent antiviral ligand for *Drosophila* STING. This discovery provides the first evidence that CDNs beyond 2'3'-cGAMP function as endogenous second messengers in metazoans.

Reviewer Reports on the First Revision:

Referees' comments:

Referee #1 (Remarks to the Author):

This revised manuscript has thoroughly addressed nearly all the previous comments. One key issue raised, by this reviewer, was the lack of genetic demonstration of cGLR1 in dsRNA sensing. Although these authors do not address this directly, the mammalian cell reconstitution assay nicely demonstrates sufficiency for cGLR1 for this response. This is satisfying, but raise an issue that was perhaps glossed over. Do these results show that hSTING can respond to 3'2'-cGAMP? Or are these cells also carrying an insect STING? This should be addressed more explicitly.

The other conundrum that is raised by new data is the activity, or not, of 2'3'-cGAMP. figure 4a appears to show no binding of this isomer to dSTING, yet clearly it has activity in vivo, albeit reduced, in inducing SRG genes and protecting against viruses. The authors need to provide an explanation for this apparent contradiction.

Referee #2 (Remarks to the Author):

The revised manuscript by Kranzusch and colleagues has improved compared to the original version. Determining the structure of 3'2'-cGAMP bound to dSTING helps understand the preferential recognition of this specific cyclic dinucleotide over the linkage isomer. The related biochemical analysis adds to the presented work. In addition, the authors attempted to strengthen the biological importance of their findings. They now demonstrate minor differences in potency between the antiviral activity of the cGAMP linkage isomers in vivo and include engineered dSTING-hSTING expressing cell line studies, but not primary cells.

Contrary to the statements in the text, the authors have not provided evidence that cGLR1 functions as a "foreign" RNA sensor in *Drosophila* immunity. Whether cGLRs present divergent receptors within a single species and the concept of "radiation" is unclear and proving it will require additional work.

Referee #3 (Remarks to the Author):

Responding to the concerns by this reviewer, the authors have provided as much as structural, biochemical and cell biological data. The structural statistics seem solid. Therefore, this reviewer agrees to promote publication of this revised paper.

Author Rebuttals to First Revision:

Referee #1

This revised manuscript has thoroughly addressed nearly all the previous comments.

One key issue raised, by this reviewer, was the lack of genetic demonstration of cGLR1 in dsRNA sensing. Although these authors do not address this directly, the mammalian cell reconstitution assay nicely demonstrates sufficiency for cGLR1 for this response. This is satisfying, but raise an issue that was perhaps glossed over. Do these results show that hSTING can respond to 3'2'-cGAMP? Or are these cells also carrying an insect STING? This should be addressed more explicitly.

We thank the reviewer for their helpful feedback. We have clarified the details of our STING constructs in the Methods section and corresponding figure legends. In Fig. 2e, we measured *Dm*-cGLR1 activation in cells using a mouse STING allele and an IFN- β reporter assay. These results demonstrate that *Dm*-cGLR1 activation is dependent on dsRNA-stimulation and indicate that mouse STING is capable of responding to 3'2'-cGAMP, consistent with previous observations that human STING can respond to 3'2'-cGAMP (Zhang et al. 2013 *Molecular Cell* PMID 23747010; Shi et al. 2015 *PNAS* PMID 26150511). In Extended Data Fig. 8d, we measured STING activation following direct delivery of 3'2'-cGAMP to permeabilized cells using a STING chimera that includes the *Drosophila* STING cyclic dinucleotide binding domain (*D. eugracilis* STING amino-acids 153–340) fused to the transmembrane and C-terminal tail regions of human STING. These results confirm that the dSTING cyclic dinucleotide binding domain preferentially responds to 3'2'-cGAMP in cells.

The other conundrum that is raised by new data is the activity, or not, of 2'3'-cGAMP. Figure 4a appears to show no binding of this isomer to dSTING, yet clearly it has activity in vivo, albeit reduced, in inducing SRG genes and protecting against viruses. The authors need to provide an explanation for this apparent contradiction.

Our results demonstrate that only 3'2'-cGAMP forms a thermo-stable complex with *Drosophila* STING *in vitro* and reveal unique contacts in the dSTING–3'2'-cGAMP crystal structure that explain specific 3'2'-cGAMP recognition (Fig. 3 and Extended Data Fig. 8). We have added a sentence to the legend of Extended Data Fig. 8b to explain that the ability of *Drosophila* STING to respond to 2'3'-cGAMP *in vivo* indicates that *Drosophila* STING is capable of weaker recognition of other cyclic dinucleotides. Our results mirror the human STING system where human STING recognizes 2'3'-cGAMP with high-affinity but is also capable of weaker recognition of other cyclic dinucleotides including bacteria-derived 3'3'-cGAMP and 3'3'-c-di-GMP (Diner et al., *Cell Reports* 2013 PMID 23707065; Ablasser et al., *Nature* 2013 PMID 23722158).

Referee #2

The revised manuscript by Kranzusch and colleagues has improved compared to the original version. Determining the structure of 3'2'-cGAMP bound to dSTING helps understand the preferential recognition of this specific cyclic dinucleotide over the linkage isomer. The related biochemical analysis adds to the presented work. In addition, the authors attempted to strengthen the biological importance of their findings. They now demonstrate minor differences in potency between the antiviral activity of the cGAMP linkage isomers in vivo and include engineered dSTING-hSTING expressing cell line studies, but not primary cells.

*Contrary to the statements in the text, the authors have not provided evidence that cGLR1 functions as a “foreign” RNA sensor in *Drosophila* immunity. Whether cGLRs present divergent receptors within a single species and the concept of “radiation” is unclear and proving it will require additional work.*

We thank the reviewer for their helpful feedback to improve our manuscript. We agree with the reviewer's points and have incorporated these text changes. See: “*Drosophila* exhibit a remarkable radiation of cGLR genes” corrected to “*Drosophila* encode a remarkable number of cGLR genes” and “*Drosophila* cGLR1 reveals a parallel signaling system for sensing foreign RNA” corrected to “*Drosophila* cGLR1 reveals a parallel signaling system for sensing dsRNA”.

Referee #3

Responding to the concerns by this reviewer, the authors have provided as much as structural, biochemical and cell biological data. The structural statistics seem solid. Therefore, this reviewer agree to promote publication of this revised paper.

We are glad the reviewer is satisfied with our revised manuscript and we thank them for their helpful feedback.